# See it to Place it: Evolving Macro Placements with Vision-Language Models

## Abstract

We propose using Vision-Language Models (VLMs) for macro placement in chip floorplanning, a complex optimization task that has recently shown promising advancements through machine learning methods. Because human designers rely heavily on spatial reasoning to arrange components on the chip canvas, we hypothesize that VLMs with strong visual reasoning abilities can effectively complement existing learning-based approaches. We introduce VeoPlace (Visual Evolutionary Optimization Placement), a novel framework that uses a VLM—without any fine-tuning—to guide the actions of a base placer by constraining them to subregions of the chip canvas. The VLM proposals are iteratively optimized through an evolutionary search strategy with respect to resulting placement quality. On open-source benchmarks, VeoPlace outperforms the best prior learning-based approach on 9 of 10 benchmarks with peak wirelength reductions exceeding 35%. We further demonstrate that VeoPlace generalizes to analytical placers, improving DREAMPlace performance on all 8 evaluated benchmarks with gains up to 4.3%. Our approach opens new possibilities for electronic design automation tools that leverage foundation models to solve complex physical design problems.

## 1. Introduction

Computer chip floorplanning is a critical step in the integrated circuit design process, involving the strategic arrangement of macros on the chip canvas. Determining the optimal placement is a complex multi-objective problem, in which performance, power, and area (PPA) must be optimized while minimizing routing congestion. The vast combinatorial design space makes manual chip floorplanning a time-consuming and expertise-driven task while posing a significant challenge for automated methods.

A variety of approaches have been proposed for automated chip floorplanning, including black-box optimization (Shi et al., 2023), analytical methods (Lin et al., 2019; Cheng et al., 2018; Lu et al., 2015), and learning-based methods (Mirhoseini et al., 2021; Lai et al., 2022; 2023; Lee et al., 2024). Among these, learning-based approaches have achieved state-of-the-art performance, but have a severe limitation: policies trained from scratch struggle to generalize to *unseen chips* without additional interaction, an issue exacerbated by the limited training data available in chip design. In contrast, human designers leverage high-level prior knowledge and spatial reasoning to efficiently tackle new design spaces. Our work aims to bridge this gap by harnessing Vision-Language Models (VLMs) to provide human-like spatial reasoning and guide the exploration of existing placement algorithms. While human designers develop placement intuition over years of experience, VLMs can rapidly extract spatial patterns from dozens of prior placements and their evaluations simultaneously—a form of in-context learning that exceeds what any individual designer can process at once.

Existing learning-based approaches first pre-train models on a set of training chips, then fine-tune on sampled placements from unseen chips (Mirhoseini et al., 2021; Lai et al., 2022; 2023). We consider a general formulation of this setup: for an unseen block and a fixed budget of $B$ placement attempts, what is the best possible placement that can be generated? We posit that efficiently using this budget of online attempts requires spatial reasoning as well as learning from prior attempts - capabilities that have been both exhibited by modern VLMs.

We introduce VeoPlace (Visual Evolutionary Optimization Placement), a novel framework that uses a high-level VLM planner to guide a low-level placer by constraining it to promising regions. Crucially, these VLM proposals are iteratively refined through an evolutionary process, as visualized in Figure 1. VeoPlace requires no fine-tuning of the VLM (we use the public Gemini models (Team et al., 2023)) and uses an independent low-level placer (e.g. ChiPFormer (Lai et al., 2023)). On open-source benchmarks (ISPD 2005 (Nam et al., 2005), ICCAD 2004 (Adya &

[1]Anonymous Institution, Anonymous City, Anonymous Region, Anonymous Country. Correspondence to: Anonymous Author <anon.email@domain.com>.

Preliminary work. Under review by the International Conference on Machine Learning (ICML). Do not distribute.

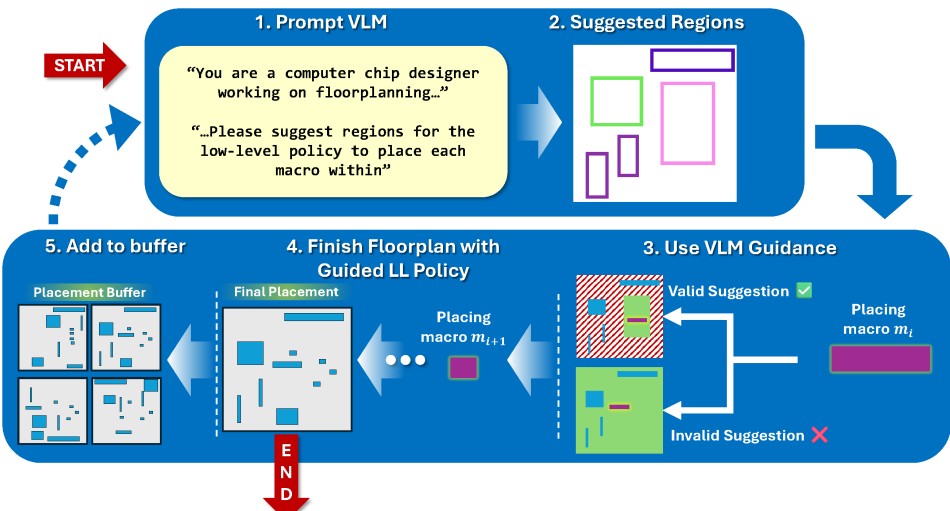

*Figure 1.* VeoPlace framework overview. The VLM suggests placement regions (1-2) to constrain a low-level placer (3) for macro placement (4). A history buffer that stores the existing population of placements (5) facilitates evolutionary in-context improvement, creating a feedback loop to improve placement quality.

Markov, 2002; Adya et al., 2004), and Ariane (OpenHW Group, 2024)), VeoPlace outperforms ChiPFormer on 9 of 10 benchmarks with peak wirelength reductions exceeding 35%. Furthermore, we show VeoPlace generalizes to analytical placers, improving DREAMPlace performance on all eight ICCAD 2015 Superblue benchmarks with gains up to 4.3%. Our main contributions are:

- **Unified VLM-guided placement framework**: We introduce VeoPlace, the first framework to show that foundation models can guide specialized placement algorithms via spatial reasoning, without requiring fine-tuning. We demonstrate two instantiations: action masking for learning-based policies and soft anchor constraints for analytical placers.

- **State-of-the-art results across placement paradigms**: VeoPlace beats DREAMPlace 4.3.0 on all eight ICCAD 2015 Superblue benchmarks (up to 4.3%) and outperforms ChiPFormer on 9 of 10 benchmarks (up to 35%), demonstrating that VLM guidance generalizes across fundamentally different placement approaches.

Practically, VeoPlace serves as a plug-in enhancement for existing placement workflows: engineers can wrap their current placer with VLM guidance to improve placement quality without retraining models or modifying their design flow.

## 2. Related Work

**Vision-Language Models for Decision-Making.** Vision-Language Models (VLMs) are trained on vast datasets of

*Table 1.* Comparison of existing macro placement approaches. Our VLM-based approach represents a novel direction in the field. RL: Reinforcement Learning, MCTS: Monte Carlo Tree Search, IL: Imitation Learning, BBO: Blackbox Optimization, VLM: Vision-Language Model, LLM: Large Language Model.

| Method | Category |
|---|---|
| SP-SA (Murata et al., 2002) | Packing |
| NTUPlace3 (Chen et al., 2008) | Analytical |
| RePlace (Cheng et al., 2018) | Analytical |
| DREAMPlace (Lin et al., 2019) | Analytical |
| GraphPlace (Mirhoseini et al., 2021) | RL |
| DeepPR (Cheng & Yan, 2021) | RL |
| MaskPlace (Lai et al., 2022) | RL |
| EfficientPlace (Geng et al., 2024) | RL + MCTS |
| ChiPFormer (Lai et al., 2023) | IL |
| WireMaskBBO (Shi et al., 2023) | BBO |
| EvoPlace (Yao et al., 2025) | LLM + Analytical |
| VeoPlace (Ours) | VLM + IL
VLM + Analytical |

text and images, and therefore contain rich priors valuable for tasks requiring both vision and language (Driess et al., 2023). The use of VLMs to perform decision-making has been explored in several fields, including robotics, where VLMs interpret natural language commands within a visual scene to guide robot actions or planning (Ahn et al., 2022; Jiang et al., 2022; Shridhar et al., 2022; Huang et al., 2022; Brohan et al., 2023; Kim et al., 2024; Team et al., 2025; Liang et al., 2024). Systems such as SayCan (Ahn et al., 2022) and RT-2 (Brohan et al., 2023) demonstrate how VLMs can translate high-level instructions into actionable plans that low-level controllers can execute.

Our work leverages these VLM capabilities in a similar hierarchical approach. The VLM perceives chip placement images along with their performance metrics, analyzes spatial arrangements, and provides high-level guidance to a low-level placer in the form of suggested bounding regions for each macro. These bounding regions constrain the low-level placer, effectively creating a division of labor where the VLM handles high-level spatial reasoning while a specialized placer executes precise placement decisions within these constraints.

**Automated Chip Floorplanning.** Automating computer chip floorplanning has been studied through various approaches (Table 1), including analytical methods (Lin et al., 2019), black-box optimization techniques such as simulated annealing (Wong & Liu, 1986) and genetic algorithms (Singha et al., 2012), more recent guided black-box methods (Shi et al., 2023), and various learning-based methods (Mirhoseini et al., 2021; Lai et al., 2023; Geng et al., 2024). Within the learning-based category, a prominent line of work formulates chip floorplanning as a reinforcement learning (RL) problem where macros are sequentially placed onto a chip canvas (Mirhoseini et al., 2021; Lai et al., 2022; 2023). Alternatively, recent works have proposed learning to refine existing chip floorplans, employing techniques such as diffusion models (Lee et al., 2024) or RL algorithms (Xue et al., 2024) for post-processing. Our approach can be viewed as a generalization of learning-based approaches (and can explicitly leverage them in the inner loop) by using a high-level VLM to guide them at test-time.

**Pairing LLMs with Evolution.** Pairing LLMs with evolutionary search has achieved success in fields such as program generation (Romera-Paredes et al., 2024; Hemberg et al., 2024; Liventsev et al., 2023), planning and reasoning (Lee et al., 2025), scientific discovery (Yamada et al., 2025; Gottweis et al., 2025), robotics (Nasiriany et al., 2024), and chip design (Novikov et al., 2025; Yao et al., 2025; Xue et al., 2024; Shi et al., 2023). Most relevant to our work, EvoPlace (Yao et al., 2025) uses LLMs to evolve the optimization *algorithm code* within an analytical placer. In contrast, VeoPlace uses a VLM to evolve the placement *solutions* directly, suggesting where macros should go based on visual reasoning over prior placements. These suggestions are evaluated, and high-performing ones are selected to inform the VLM's next generation of proposals, given feedback from an objective function. Our selection strategy samples a population of high-performing, geometrically similar placements. That is, VeoPlace explicitly focuses on evolution in a local region, a principle that has been shown to be effective in sparse Gaussian processes (Wei et al., 2024) and island models in genetic algorithms (Romera-Paredes et al., 2024; Lee et al., 2025; Tanese, 1989; Cantú-Paz et al., 1998).

# 3. Preliminaries

**Macro Placement.** We consider macro placement in chip floorplanning, where a set of *macros* $M = \{m_1, \ldots, m_N\}$, defined by their dimensions and connectivity, are placed on a 2D chip *canvas*. Connectivity is given by a *netlist* $G = (M, E)$, a hypergraph where each hyperedge (*net*) connects a subset of macros. The objective is to find a placement $P = \{p_1, \ldots, p_N\}$, where $p_i$ is the bottom-left corner of macro $m_i$, that minimizes estimated wirelength. This is the total length of wiring needed to connect the components (macros and standard cells) within each net, and is a crucial metric for a chip's performance, power, and area (PPA) (Lin et al., 2019; Mirhoseini et al., 2021).

Macro placement is formulated as a sequential decision-making problem, a Markov Decision Process (MDP) (Mirhoseini et al., 2021; Lai et al., 2023; 2022). In this setup, macros are sequentially placed onto the canvas, typically following a predefined order such as descending macro area. The state $s_t$ encompasses information about the current partial placement (locations of macros $m_1, \ldots, m_{t-1}$), features of the current macro $m_t$, and potentially structural information derived from the netlist $G$.

To manage the continuous placement space, the canvas is discretized into a grid of cells, where an action $a_t$ selects a specific grid cell for the reference point (e.g., the bottom-left corner) of the current macro, $m_t$. After all $N$ macros are placed, a terminal reward $R$ is computed based on the final Half-Perimeter Wirelength (HPWL). The total HPWL is the sum of the half-perimeters of the smallest axis-aligned bounding box for each net in the netlist $G$. The agent's goal is to learn a policy $\pi(a_t|s_t)$ that maximizes the expected terminal reward $\mathbb{E}[R]$ (or, equivalently, minimizes HPWL).

**Inference-time Optimization.** Online RL requires many environment interactions and model updates to produce optimal placements for new netlists. Recent work suggests that offline RL pre-training provides strong zero-shot performance but benefits from fine-tuning on a small amount of online interaction (Lai et al., 2023). We consider an inference-time optimization setting that uses a **hierarchical approach**: we are allowed a fixed budget of placement evaluations, but do not fine-tune either the VLM (our high-level strategic guide) or the low-level placer. As Section 5 shows, VeoPlace can achieve results superior to fine-tuning, suggesting greater efficiency on new tasks.

**Analytical Placer.** Analytical placers formulate placement as a nonlinear optimization problem, minimizing wirelength subject to density constraints. We use DREAM-Place (Lin et al., 2019), a state-of-the-art GPU-accelerated analytical placer. In our framework, we guide DREAM-

Place by incorporating VLM suggestions as soft anchor constraints in the optimization objective, pulling macros toward suggested target locations while still allowing the solver to find globally optimal arrangements.

**Learning-Based Placer.** ChiPFormer (Lai et al., 2023) is an autoregressive Transformer that formulates macro placement as offline reinforcement learning via a Decision Transformer objective (Chen et al., 2021; Lee et al., 2022), achieving state-of-the-art performance across multiple chip designs. We adopt it as our learning-based low-level placer due to its multi-task generality. Crucially, ChiPFormer outputs a probability distribution over grid cells for each macro. In our framework, the VLM constrains this distribution by masking out regions outside its suggested placement areas, steering the policy toward better design choices without requiring any fine-tuning.

## 4. Method

In this section, we describe VeoPlace, our novel evolutionary framework that harnesses the spatial reasoning of VLMs for chip floorplanning. The framework iteratively evolves a population of placements, using a VLM as a variation operator. The VLM generates region proposals based on prior attempts, providing spatial guidance to a low-level placer—either as soft anchor constraints for analytical placers or action masks for learning-based policies. As illustrated in Figure 1 and Algorithm 1, we build a context from the history buffer and query the VLM for region suggestions (lines 3–5), then use these suggestions to constrain the low-level placer as it generates a complete floorplan (line 9). High-performing placements are stored in the history buffer (line 12), creating an evolutionary feedback loop that continuously improves placement quality.

### 4.1. VLM and Low-Level Placer Interface

The manner in which VLM suggestions are incorporated depends on the low-level placer: for analytical placers, suggestions become soft anchor constraints in the optimization objective; for learning-based policies, suggestions mask the action space to constrain sampling. We demonstrate VeoPlace with both approaches. Algorithm 1 summarizes the overall procedure, while PLACEMACROS is instantiated for analytical and learning-based placers in Algorithms 2 and 3, respectively. In both paradigms, we alternate between unguided and VLM-guided rollouts: 8 episodes sample the base placer directly, then 8 episodes use VLM guidance. Since the VLM API returns 8 candidate suggestions per query, we require only one API call every other iteration—for 2,000 rollouts, amounting to just 125 API calls. After each full placement, we legalize placement $P$ using DREAMPlace's built-in legalizer to ensure a valid macro

---

**Algorithm 1 VeoPlace**

**Require:** V: VLM; $\mathcal{P}$: Low-level placer; G: Netlist
    $C$: Context length; $E$: Episodes; $K$: VLM query interval
1: Initialize population $H \leftarrow \emptyset$
2: **for** $e = 1$ to $E$ **do**
3:    **if** $e \bmod K = 0$ **then**
4:       context $\leftarrow$ BUILDCONTEXT$(H, C)$
5:       $S \leftarrow V(\text{context}, G)$ {VLM suggestions}
6:    **else**
7:       $S \leftarrow \emptyset$
8:    **end if**
9:    $P_e \leftarrow$ PLACEMACROS$(G, S)$
10:    $P_e \leftarrow$ LEGALIZE$(P_e)$
11:    $H \leftarrow H \cup \{(P_e, \text{HPWL}(P_e))\}$
12: **end for**
13: **return** $H$

---

**Algorithm 2 PLACEMACROS (Analytical)**

**Require:** G: Netlist; S: VLM suggestions (possibly empty)
1: **if** $S = \emptyset$ **then**
2:    Solve standard DREAMPlace objective to obtain $P$
3: **else**
4:    Convert each $s_i \in S$ to anchor $\hat{x}_i$
5:    Solve anchored objective with $A$ to obtain $P$
6: **end if**
7: **return** $P$

---

**Algorithm 3 PLACEMACROS (Learned)**

**Require:** $\pi$: Policy; G: Netlist; S: VLM suggestions
1: Order macros by connectivity, size
2: Initialize $P \leftarrow \emptyset$
3: **for** macro $m_t$ **do**
4:    **if** $S$ provides valid suggestion $s_t$ **then**
5:       $p_t \sim \pi(\cdot | m_t, P, s_t)$
6:    **else**
7:       $p_t \sim \pi(\cdot | m_t, P)$
8:    **end if**
9:    $P \leftarrow P \cup \{(m_t, p_t)\}$
10: **end for**
11: **return** $P$

---

and standard-cell placement.

### 4.1.1. GUIDING ANALYTICAL PLACERS

For analytical placers, we interpret VLM suggestions as target macro locations. Let $x$ denote macro locations, $y$ denote standard-cell locations, $E$ denote the set of nets, and $M$ denote the set of macros. The standard DREAMPlace objective is

$$\min_{x,y} \sum_{e \in E} \text{WL}(e; x, y) + \lambda D(x, y), \quad (1)$$

where $D$ is the density penalty. We incorporate VLM suggestions by adding an anchor term with weight $\lambda_A$:

$$\min_{x,y} \sum_{e \in E} \text{WL}(e; x, y) + \lambda D(x, y) + \lambda_A A(x; \{\hat{x}_i\}), \quad (2)$$

where $A$ penalizes distance from each macro $i$ to its suggested location $\hat{x}_i$. We convert the VLM's bounding box suggestion to an anchor point by taking the bottom-left

corner of the region as $\hat{x}_i$. We use a quadratic anchor:

$$A(x; \{\hat{x}_i\}) = \sum_{i \in M} \|x_i - \hat{x}_i\|_2^2, \qquad (3)$$

which pulls macros toward the suggested targets. The weight $\lambda_A$ controls the strength of VLM guidance, where $\lambda_A = 0$ recovers unconstrained DREAMPlace. DREAMPlace solves this anchored optimization where macros are softly pulled toward VLM targets (visualized in Figure 2).

### 4.1.2. GUIDING LEARNING-BASED PLACERS

For learning-based policies, we use a stochastic low-level placement policy, $\pi$, that parameterizes a probability distribution over grid locations to place the next macro. For learning-based policies (Algorithm 3), macros are placed sequentially, ordered by connectivity then size, ensuring that the largest and most highly connected macros are placed first. The VLM suggests bounding box regions $\{s_1, ..., s_N\}$ on the chip canvas for the respective macros (visualized in Figure 5).

VeoPlace then rolls out the low-level policy, but constrains its actions at each timestep $t$ to the suggested region $s_t$, denoted by $\pi(\cdot|m_t, P_e, s_t)$ (Algorithm 3). This is practically achieved by masking the policy's output logits outside of $s_t$ before sampling. This constrains $\pi$ to place $m_t$ within the area identified as promising by the VLM, while still retaining control over the exact placement coordinates within the region.

Because the low-level policy autoregressively places macros, a VLM suggestion $s_t$ may be invalid for any $t > 1$ due to already-placed macros overlapping the region. In this case, the macro $m_t$ is placed by sampling from the original, unconstrained policy distribution, $p_t \sim \pi(\cdot|m_t, P_e)$. Empirically, this fallback is triggered for $\sim 20\%$ of suggestions once evolution stabilizes (Figure 4d). Finally, after each placement $P_e$ is completed, its quality (e.g., $HPWL_e$) is calculated and the pair $(P_e, HPWL_e)$ is added to the population $H$.

### 4.2. Structured Prompt

VeoPlace prompts a VLM to generate bounding box suggestions for each macro $\{s_1, ..., s_N\}$ conditioned on previous placements and their evaluations $\{P_i\}$. We generate suggestions for all macros simultaneously to reduce VLM inference. The prompt's key characteristics are (1) its structure, which elicits spatial reasoning, and (2) the selection of in-context examples (detailed in Section 4.3, with a full example in Appendix E.1).

We find that VLMs struggle with macro placement due to information overload and a lack of domain-specific knowledge, often producing inconsistent or imprecise spatial sug-

gestions without proper guidance (see Appendix E.3). Our structured prompt guides the VLM with clear objectives, constraints, and a standardized format (Table 2), transforming its general visual reasoning into useful spatial guidance. To ensure generalizability and avoid overfitting, we developed the structure and syntax of our prompt exclusively on the adaptec1 benchmark to ensure valid spatial reasoning, applying the final version to all other benchmarks without modification.

*Table 2.* Components of the structured prompt for VeoPlace.

| Component | Description |
|---|---|
| **Grid Repr.** | • 84×84 grid for ChiPFormer experiments (matching (Lai et al., 2023)); 512×512 grid for DREAMPlace experiments.
• Macros positioned by bottom-left corner coordinates.
• Origin (0,0) at the bottom-left. |
| **Visual Repr.** | • Image of canvas showing placed macros and colors.
• Visual context for spatial relationships and patterns. |
| **Context** | • Grid specs and macro properties (dimensions, color).
• History of prior placements with performance metrics.
• Current state (locations) of placed macros. |

### 4.3. Context Selection Strategies for Evolution

The core component of our evolutionary algorithm is prompting the VLM to generate a superior placement suggestion given a set of prior placements and their evaluations. Because each placement uses hundreds of tokens, only a small number can be provided to the model while maintaining reasonable inference cost. Given this limited budget, the examples should be (1) high-quality, so the model improves upon good placements, and (2) informative enough for the model to effectively deduce better placements via reasoning.

We compare multiple context selection strategies (FIFO, Random, Best, Diverse, Top Stratified) in Section 5.3. We use **Top Stratified** as our default because it balances exploration (sampling across clusters) with exploitation (favoring high-performing placements within a promising cluster).

## 5. Experiments

We design experiments to address two key questions: **(Q1)** Can VLM guidance improve the placement quality of both learning-based policies and analytical placers using only inference-time computation? **(Q2)** Which VeoPlace design choices (anchor weight, context selection strategy, and con-

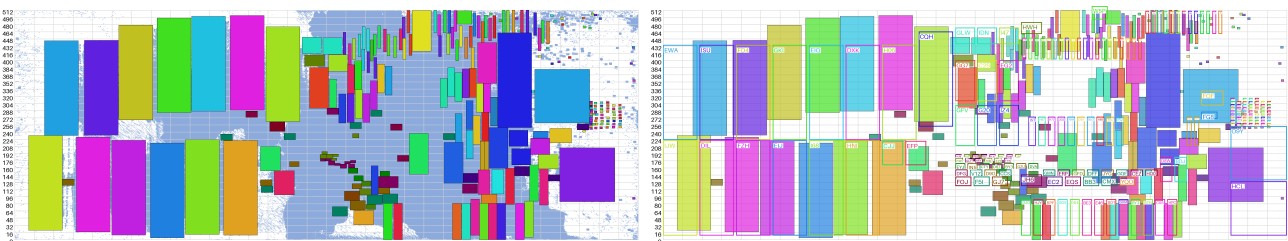

*Figure 2*. VLM guidance visualization on `superblue1`. **Left:** Final macro placement produced by VeoPlace + DREAMPlace 4.3.0. **Right:** The same placement with VLM-suggested regions overlaid. The VLM proposes target positions for each macro, and DREAMPlace's loss function is modified with anchor weights to keep macros close to VLM suggestions while allowing standard cells to optimize freely (standard cells removed for visual clarity).

text length) most affect performance?

## 5.1. Experimental Setup

For both experimental tracks, VeoPlace generates an equal number of guided and unguided rollouts in each batch: 8 episodes from directly sampling the base placer, and 8 episodes guided by VLM suggestions (see Appendix Table 6 for Gemini sampling parameters). For all benchmarks, the VLM provides suggestions for at most 256 macros, matching the maximum number of macros placed by ChiPFormer (see Table 5 for benchmark statistics).

### 5.1.1. ANALYTICAL PLACER

We evaluate on the ICCAD 2015 Superblue benchmarks (Kim et al., 2015) using DREAMPlace 4.3.0 (Lin et al., 2019) as the base placer. As described in Section 4.1.1, VLM suggestions are incorporated as soft anchor constraints in the DREAMPlace optimization objective. For DREAMPlace, each rollout corresponds to a run with a different random seed/initialization.

We discretize the canvas to a $512 \times 512$ grid for VLM suggestions and use Gemini 2.5 Flash with $C=25$ context examples. All experiments run 2,000 rollouts across three seeds and report global HPWL (see Section C.3.1 for full hyperparameters).

### 5.1.2. LEARNING-BASED PLACER

We evaluate on open-source chip benchmarks from the ISPD 2005 challenge (Nam et al., 2005) and ICCAD 2004 (Adya & Markov, 2002; Adya et al., 2004), following (Lai et al., 2023), and the Ariane RISC-V CPU (OpenHW Group, 2024). These benchmarks vary in complexity, with hundreds to thousands of macros and up to hundreds of thousands of standard cells. We train ChiPFormer from scratch using the public repository[1] and use an $84 \times 84$ grid to match its original setup. We use Gemini 2.5 Flash with $C=1$, as even a single in-context example is enough to guide ChiP-

---

[1] https://github.com/laiyao1/chipformer

Former to better solutions. We run 2,000 rollouts across three seeds (further details in Appendix C.2.2; see Figure 5 for a visualization of VLM-guided placement).

Our setup differs from (Lai et al., 2023) in two aspects. (1) During the evolutionary search, we cluster standard cells for faster reward computation, following prior work (Mirhoseini et al., 2021; Lee et al., 2024), but report unclustered results for final evaluation. (2) During final evaluation, macro locations are fixed and DREAMPlace is used only to place standard cells around them, matching (Mirhoseini et al., 2021). Empirically, allowing movable macros results in significant changes to their final placements, confounding the actual efficacy of the base placer (see Figure 7).

## 5.2. Q1: Does VLM Guidance Improve Placement Quality?

**Analytical Placers.** As shown in Table 3, VeoPlace outperforms DREAMPlace 4.3.0 on all eight Superblue benchmarks, with improvements ranging from 0.7% (superblue1) to 4.3% (superblue7). Figure 3 visualizes example placements comparing VeoPlace to DREAMPlace. We also report a congestion proxy (RUDY) in Table 11; differences are minor, suggesting that VeoPlace improves wirelength without negatively affecting routability. We analyze the sensitivity to anchor weight, context selection strategy, and context length in Section 5.3.

**Learning-Based Placers.** Our evaluation against ChiP-Former baselines demonstrates that VeoPlace consistently improves placement quality. We compare against two strong baselines under the same 2,000-rollout budget: a version without fine-tuning (No-FT) that repeatedly samples the fixed ChiPFormer policy, and a version with ChiPFormer fine-tuning (FT) that adapts the policy via online Decision Transformer. As detailed in Table 4, VeoPlace outperforms ChiPFormer on 9 of 10 benchmarks, with pronounced gains on `ibm2` (32%) and `ariane136` (26%).

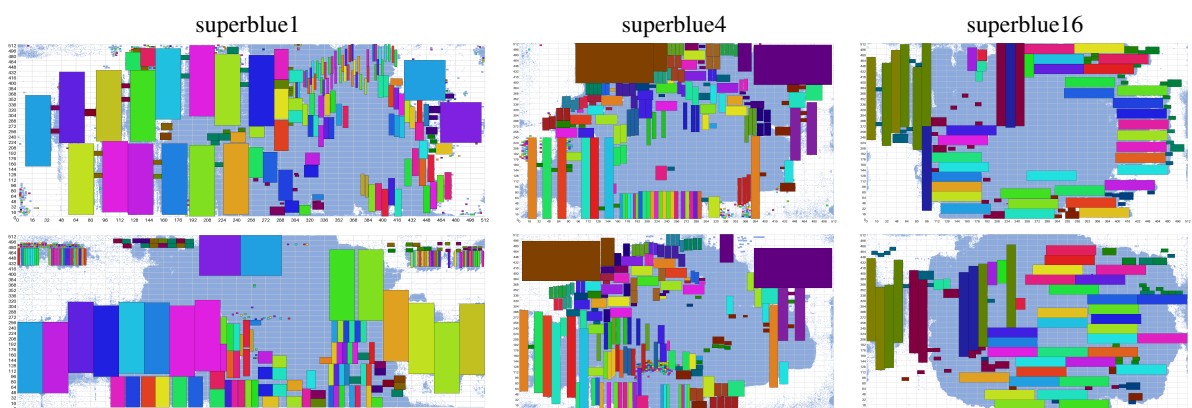

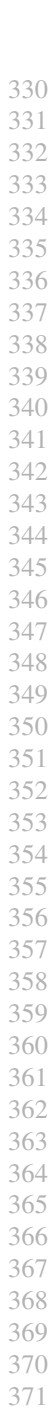

*Figure 3.* Visual comparison of placements on selected Superblue benchmarks. **Top row:** DREAMPlace 4.3.0. **Bottom row:** VeoPlace-guided DREAMPlace. Blue clouds are individual standard cells; colored rectangles are macros. VeoPlace guidance improves global HPWL on all eight superblue benchmarks.

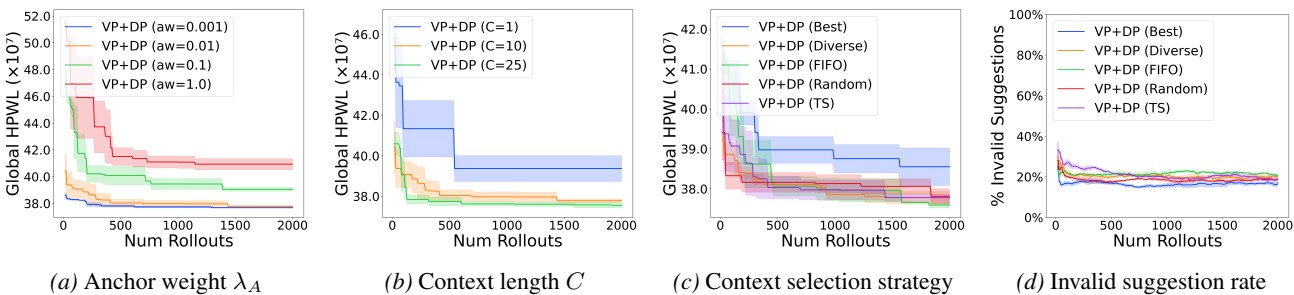

*(a)* Anchor weight $\lambda_A$     *(b)* Context length $C$     *(c)* Context selection strategy     *(d)* Invalid suggestion rate

*Figure 4.* Design choice ablations on `superblue1` with VeoPlace guiding DREAMPlace 4.3.0. Unless varied, defaults are: top stratified (TS) strategy, $C$=10, $\lambda_A$=0.01. (a) Lower anchor weights perform best, giving the analytical placer freedom to optimize standard cells around macros. (b) Longer context ($C = 25$) achieves lower HPWL. (c) All strategies improve with further rollouts. (d) For all strategies, invalid suggestion rate converges to ~20%.

*Table 3.* VeoPlace-guided DREAMPlace vs. DREAMPlace 4.3.0 on Superblue. VP+DP = VeoPlace with DREAMPlace; DP = DREAMPlace; 2.5 Flash = Gemini 2.5 Flash. We report mean and standard error of best global HPWL ($\times 10^7$, lower is better) across three random seeds with 2,000 rollouts each.

| Benchmark | VP+DP 2.5 Flash ($C$=25) | DP 4.3.0 |
|---|---|---|
| superblue1 | **37.56±0.19** | 37.81±0.04 |
| superblue3 | **42.20±0.05** | 43.13±0.17 |
| superblue4 | **28.85±0.33** | 29.24±0.02 |
| superblue5 | **39.53±0.20** | 40.10±0.11 |
| superblue7 | **54.64±0.31** | 57.12±0.20 |
| superblue10 | **67.84±0.04** | 68.72±0.08 |
| superblue16 | **36.51±0.03** | 36.99±0.12 |
| superblue18 | **21.54±0.14** | 22.17±0.05 |

### 5.3. Q2: Which design choices matter?

Beyond the VLM itself, VeoPlace is parameterized by several design choices. We study three key factors on the `superblue1` benchmark (Figure 4): anchor weight $\lambda_A$, context selection strategy, and context length $C$. We also ablate prompt strategy on `adaptec1`.

**Prompt Strategy.** We test two prompt variants: a **greedy** prompt that instructs the VLM to make only minor modifications to existing placements, and an **exploration-focused** prompt that encourages novel placements. The exploration-focused prompt consistently outperforms the greedy variant (Table 12), confirming that exploiting known solutions alone is insufficient.

**Anchor Weight.** The anchor weight $\lambda_A$ controls how strongly DREAMPlace is guided toward VLM-suggested positions. As shown in Figure 4a, lower weights perform better, with $\lambda_A = 0.001$ achieving the lowest global HPWL, while $\lambda_A \in \{0.1, 1.0\}$ degrades performance. Lower anchor weights allow standard cells to flow freely around the VLM suggested macro positions, whereas higher weights overly constrain the optimization (see Section A.2.2 for a qualitative comparison).

**Context Selection Strategy.** We compare five strategies for selecting the $C$ in-context examples from the history buffer:

**Most Recent (FIFO)**: Select the $C$ most recently generated

*Table 4.* Comparison of VeoPlace (VP) and ChiPFormer (CF). VP+CF = VeoPlace with ChiPFormer; 2.5 Flash = Gemini 2.5 Flash; No-FT/FT = without/with ChiPFormer fine-tuning. We report mean global HPWL ($\times 10^7$, lower is better) with standard error across three random seeds over 2,000 rollouts. The best result for each benchmark is bolded.

| Benchmark | VP+CF 2.5 Flash ($C = 1$) | ChiPFormer | |
| --- | --- | --- | --- |
| | | No-FT | FT |
| adaptec1 | 15.97±0.80 | **14.23±0.77** | 14.99±0.55 |
| adaptec2 | **11.82±0.89** | 15.03±0.59 | 13.71±0.33 |
| adaptec3 | **24.10±0.21** | 25.64±0.43 | 26.31±0.97 |
| adaptec4 | **19.63±0.20** | 22.65±1.69 | 21.50±0.50 |
| ibm1 | **0.31±0.01** | 0.44±0.04 | 0.40±0.01 |
| ibm2 | **0.59±0.01** | 0.91±0.02 | 0.87±0.06 |
| ibm3 | **0.80±0.02** | 1.16±0.10 | 0.96±0.02 |
| ibm4 | **0.84±0.00** | 1.15±0.06 | 0.96±0.02 |
| ariane133 | **0.34±0.02** | 0.46±0.00 | 0.45±0.02 |
| ariane136 | **0.34±0.02** | 0.46±0.01 | 0.48±0.01 |

placements from the population. This strategy implements pure evolutionary search, where the VLM observes a temporally ordered sequence of recent attempts.

**Random**: Randomly sample $C$ placements from the population buffer. This provides a baseline that makes no assumptions about which examples are most valuable for evolutionary search.

**Best Performing**: Select the $C$ placements with the lowest global HPWL from the population to encourage the VLM to replicate successful patterns.

**Diverse**: Represent each placement as a vector in $\mathbb{R}^{2T}$ by concatenating the $(x_i, y_i)$ coordinates of its $T$ macros. We then perform K-means clustering with $C$ clusters on the population and select the placement with the best HPWL from each cluster. This promotes geometric diversity across the selected examples while still favoring high-quality designs.

**Top Stratified**: Represent placements as coordinate vectors and cluster them by geometric similarity, as in the diverse strategy. We then focus on a single promising cluster by ranking all clusters by their best global HPWL and sampling one using a softmax distribution over these ranks (with probability proportional to $e^{-i/\tau}$ for rank $i$). From the selected cluster, we choose the top $C$ performing layouts, supplementing from nearby clusters if needed. This strategy provides a set of geometrically similar examples that represent variations of a particular design pattern.

As shown in Figure 4c, each strategy continues to improve over rollouts, suggesting VeoPlace is robust to the choice of strategy. Notably, "Best" performs worst, indicating that greedily selecting high-performing placements causes the evolutionary loop to get stuck in local minima.

**Invalid Suggestion Rate.** The VLM does not know which low-level placer is being used. When it suggests a region that conflicts with already-placed macros, the suggestion is discarded: for learning-based placers, the policy is left unconstrained; for analytical placers, the corresponding anchor term is removed from the loss function. As shown in Figure 4d, all strategies converge to ~20% invalid suggestions, indicating that the VLM learns to propose feasible regions as evolution progresses.

**Context Length.** We test $C \in \{1, 10, 25\}$ examples. As shown in Figure 4b, $C = 25$ achieves lower global HPWL, suggesting that more in-context examples help the VLM better understand patterns associated with high-quality macro placements.

## 6. Limitations & Future Work

Our evaluation focuses on wirelength (global HPWL) as the primary optimization objective. While wirelength is a well-established proxy for circuit quality, high-quality chips require optimizing other important PPA metrics like timing and power. However, because VeoPlace relies on natural language prompting rather than a fixed loss function, additional PPA metrics can be included directly in the VLM's prompt, allowing it to reason about trade-offs that are difficult to encode mathematically. Additionally, VeoPlace relies on VLM API calls, introducing latency overhead from VLM inference. This overhead could be addressed by distilling the VLM's placement strategies into a smaller, open-source model. We view extending VeoPlace to multi-objective PPA optimization and reducing inference cost as promising directions for future work.

## 7. Conclusion

We proposed VeoPlace, an evolutionary framework that leverages vision-language models to enhance chip floorplanning. Through a structured prompting approach that requires no fine-tuning of the VLM, VeoPlace exceeds the performance of state-of-the-art learning-based methods and analytical placers, providing a blueprint for integrating VLMs into complex engineering workflows as high-level strategic guides that pave the way for more accessible and powerful computer-aided design.

## Impact Statement

This work applies vision-language models to chip floorplanning, an established step in integrated circuit design. We do not foresee specific negative societal consequences beyond those generally associated with advances in electronic design automation and machine learning.

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

# Appendix

# A. Additional Details

## A.1. Benchmark Statistics

*Table 5.* Benchmark statistics for all evaluated circuits. IBM benchmarks do not distinguish macros from standard cells.

| Benchmark | # Objects | # Macros | # Nets | # Pins |
|---|---|---|---|---|
| *ISPD 2005* | | | | |
| adaptec1 | 211K | 63 | 221K | 944K |
| adaptec2 | 255K | 127 | 266K | 1.1M |
| adaptec3 | 452K | 58 | 467K | 1.9M |
| adaptec4 | 496K | 69 | 516K | 1.9M |
| *ICCAD 2004* | | | | |
| ibm01 | 13K | — | 14K | 51K |
| ibm02 | 20K | — | 20K | 81K |
| ibm03 | 23K | — | 27K | 94K |
| ibm04 | 28K | — | 32K | 106K |
| *Ariane (Nangate45)* | | | | |
| ariane133 | — | 133 | — | — |
| ariane136 | — | 136 | — | — |
| *ICCAD 2015 Superblue* | | | | |
| superblue1 | 1,215K | 424 | 1,216K | 3.8M |
| superblue3 | 1,219K | 565 | 1,225K | 3.9M |
| superblue4 | 802K | 300 | 803K | 2.5M |
| superblue5 | 1,090K | 770 | 1,101K | 3.2M |
| superblue7 | 1,938K | 441 | 1,934K | 6.4M |
| superblue10 | 984K | 1,629 | 1,898K | 5.6M |
| superblue16 | 986K | 99 | 1,000K | 3.0M |
| superblue18 | 772K | 201 | 772K | 2.6M |

## A.2. Visual Examples

### A.2.1. VLM-GUIDED PLACEMENT

To provide a qualitative illustration of our method, Figure 5 visualizes a complete VLM-guided rollout on the `adaptec4` benchmark. This example highlights the hierarchical division of labor central to VeoPlace: the VLM provides a high-level spatial strategy, and the low-level policy executes the precise placement decisions within that strategy. The figure shows the initial VLM proposals ($t = 0$), the state mid-placement ($t = T/2$), and the final floorplan ($t = T$) that results from this guided process.

### A.2.2. ANCHOR WEIGHT COMPARISON

To illustrate the effect of anchor weight on placement quality, Figure 6 shows the final placements for `superblue1` at different anchor weights. With low anchor weights ($\lambda_A = 0.01$), DREAMPlace has sufficient flexibility to route standard cells around the VLM-suggested macro positions. Higher anchor weights ($\lambda_A = 1.0$) overly constrain the optimization, resulting in suboptimal standard cell placement.

## A.3. Macro Coloring

We employ a color-coding strategy to implicitly convey functional relationships between macros to the VLM. This process involves several steps:

First, we construct a **macro-connectivity graph** from the original netlist $G$. In this graph, nodes (excluding standard cells) represent the macros to be placed. An undirected, weighted edge is created between any two macros if they share one or more nets in $G$. The weight of such an edge is proportional to the number of nets these two macros commonly share. This construction effectively flattens the hypergraph structure of the netlist into a standard graph, where indirect connections through nets are represented as weighted between macros.

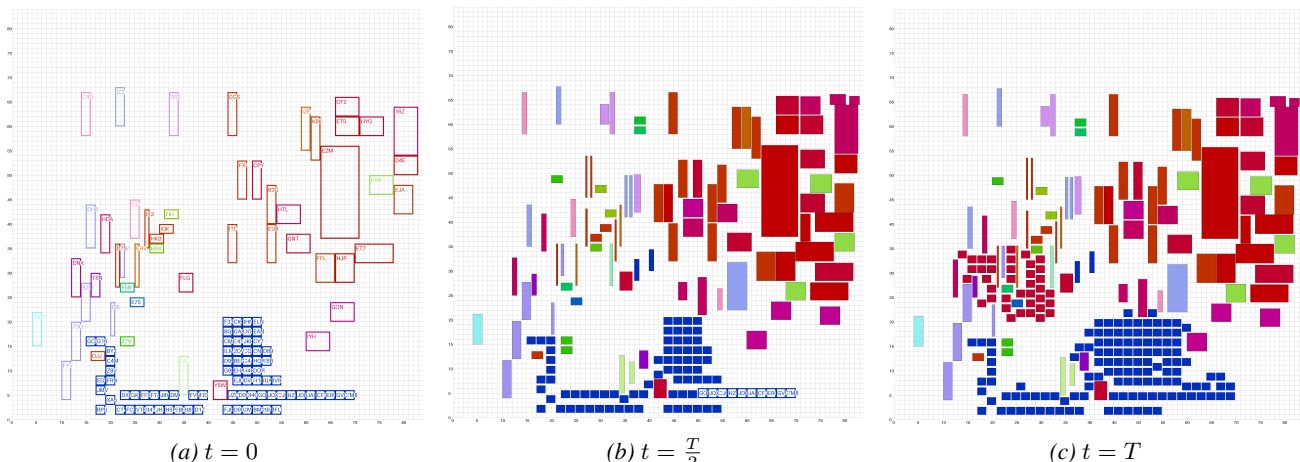

*Figure 5.* VeoPlace's VLM-guided placement on adaptec4. (a) VLM proposes initial regions ($t = 0$); policy is unconstrained for macros without valid suggestions. (b) Mid-placement ($t = T/2$). (c) Final placement ($t = T$), with the policy operating within VLM constraints.

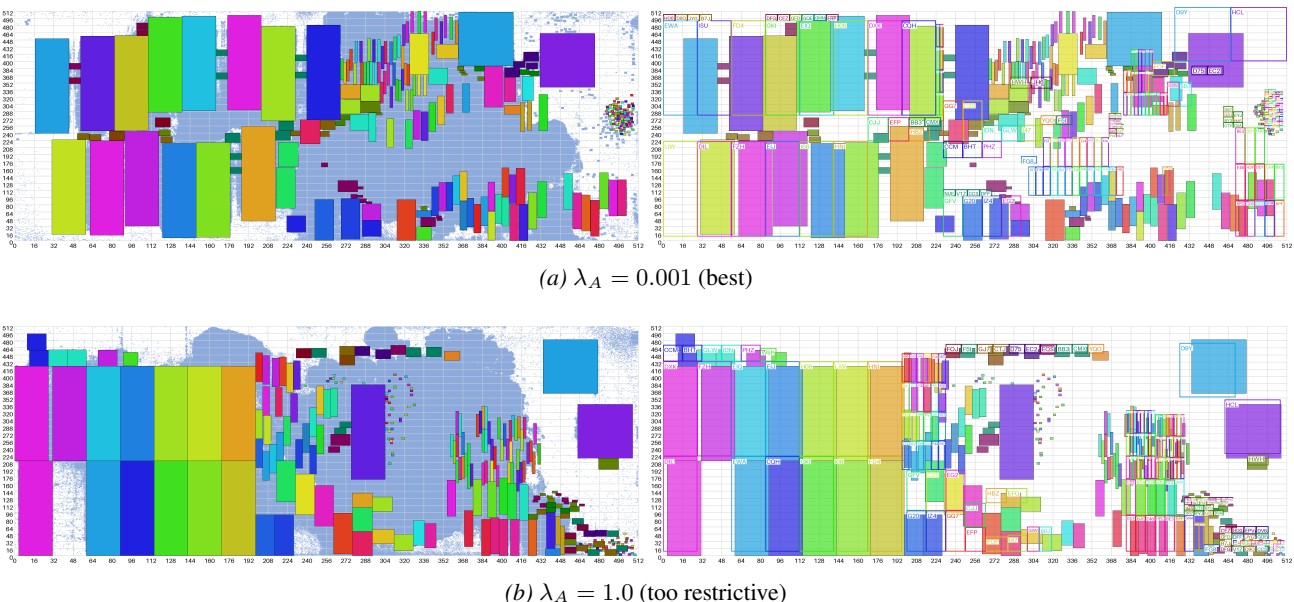

*Figure 6.* Visual comparison of placements at different anchor weights on superblue1. Lower anchor weights allow the analytical placer more flexibility to optimize standard cell positions around the macro placements.

This macro-connectivity graph is embedded into a low-dimensional space (specifically, an 8-dimensional space in our implementation for k-means) using a spring-based graph layout algorithm. Such algorithms, like the one implemented in the NetworkX (Hagberg et al., 2008) Python library, position macros in the embedding space such that those with stronger connections in the graph are located closer to one another in space.

With macros represented as points in this embedding space, we apply k-means clustering to group them. To determine a suitable number of clusters, $k$, we iterate through a predefined range of potential $k$ values (e.g., from 2 to 30). For each $k$, we perform k-means clustering and evaluate the resulting cluster separation using the Silhouette score (Rousseeuw, 1987). The value of $k$ (and its corresponding clustering) that yields the highest Silhouette score is selected as optimal.

Finally, macros are assigned colors based on their cluster membership: all macros within the same cluster receive the same unique color. Any macros that are not part of the main connectivity graph (e.g., isolated macros not sharing nets with other considered macros, if any) are assigned a default gray color. While specific netlist connectivity details are not directly fed to the VLM, this color-coding, derived from the underlying circuit structure, provides a strong visual heuristic for potential functional groupings and spatial affinities.

## B. Experimental Setup

### B.1. Justification for Fixed Macros During Standard Cell Placement

This section applies to the **learning-based** evaluation only (Section 5.1.2). For the VLM-guided analytical placer (Section 5.1.1), DREAMPlace directly optimizes macro and standard cell positions jointly via soft anchor constraints, so this concern does not arise.

In the learning-based evaluation, we report results with fixed macro placements, where the analytical placer only positions the standard cells around the macros (a two-stage flow). This differs from some prior work that allows the analytical placer to move all cells, including macros (a three-stage flow). We find that allowing macros to be movable during the analytical placement stage results in substantial displacements from their original VeoPlace-generated locations. This confounds the evaluation, as the final placement is no longer representative of VeoPlace's learned spatial reasoning but rather the output of the analytical placer.

Figure 7 provides a clear visual demonstration of this effect on the `adaptec1` benchmark. The significant displacement of macros is evident, justifying our decision to use a fixed-macro evaluation to ensure we are measuring the direct efficacy of VeoPlace.

*(a)* Our two-stage flow (VeoPlace with fixed macros).      *(b)* Alternative three-stage flow (movable macros).

*Figure 7.* Visual comparison demonstrating why a fixed-macro flow is essential for a fair evaluation of VeoPlace. (a) The final layout from our two-stage flow, where macro placements generated by VeoPlace are **fixed**. (b) The result of an alternative three-stage flow, which takes the **exact same initial placement** from (a) as input but allows DREAMPlace to move all cells. The analytical placer's drastic rearrangement of macros in (b) shows that it effectively ignores the initial solution, making its final metrics an invalid measure of VeoPlace's contribution. Our fixed-macro approach ensures a direct and unconfounded evaluation.

### B.2. Model Size

The ChiPFormer decision transformer model contains approximately 3 million trainable parameters.

### B.3. Pre-training

The pre-training phase was conducted on servers equipped with 4× NVIDIA A100 40GB GPUs.

### B.4. Rollout and Inference

For generating rollouts, the computational requirements were significantly lower as these involved only forward passes through the trained 3M parameter model. These were conducted on a single A100 GPU or equivalent, with each benchmark circuit rollout typically completing within seconds.

### B.5. VLM Integration

For VLM integration, we used Google's Gemini API with the Gemini 2.5 Flash model. Our experiments are organized into iterations, where each iteration consists of 8 rollouts. These iterations alternate: one iteration is generated using only the low-level placer, and the next is generated with VLM guidance.

A VLM query is performed once at the beginning of each guided iteration. Over a 2,000-rollout experiment, this results in 250 total iterations (125 unguided and 125 guided), leading to exactly 125 calls to the VLM. Each call to Gemini returns 8 candidate generations, with each generation being a complete set of suggested regions for all macros in the netlist.

## C. Hyperparameters

### C.1. Gemini

We use the public Gemini API endpoint for our experiments. Table 6 shows these hyperparameters.

*Table 6.* Gemini API hyperparameters.

| Parameter | Value |
|-----------|-------|
| Temperature | 0.7 |
| Top-$k$ | 64 |
| Top-$p$ | 0.95 |
| Candidates | 8 |

## C.2. Standard Cell Grouping Parameters

As discussed in Section 5, for the learning-based benchmarks (adaptec/bigblue/ibm/ariane), we group the hundreds of thousands of standard cells into a smaller set of clusters to make training and inference tractable. This grouping is only used to compute the proxy wirelength reward during rollouts; all reported results are global HPWL computed by DREAMPlace with fixed macros.

We use the open-source codebase from Google's Circuit Training project for this task, which implements the grouping methodology first introduced by Mirhoseini et al. (2021). To ensure a consistent basis for comparison, we applied the same set of grouping hyperparameters across all learning-based benchmarks. The specific values for these parameters are detailed in Table 7.

*Table 7.* Hyperparameters for the Standard Cell Grouping Algorithm.

| Parameter | Value | Description |
|-----------|-------|-------------|
| Number of Groups | 2000 | The fixed number of clusters to group all standard cells into. |
| Cell Area Utilization | 1.25 | A target for the density of cells within a cluster. |
| Enable Group Breakup | True | A boolean flag that allows the algorithm to split larger groups. |

### C.2.1. PRETRAINING

**Circuit Tokens** For pretraining the circuit token representation component using the Variational Graph Auto-Encoder (VGAE), we used the following hyperparameters:

- Hidden layer dimensions: [32, 32]

- Learning rate: 0.01

- Training epochs: 800

**Transformer** Following ChiPFormer (Lai et al., 2023), we use a reward-conditioned transformer with the following hyperparameters:

- Number of transformer layers: 6

- Number of attention heads: 8

- Embedding dimension: 128

### C.2.2. ROLLOUT SETTINGS

**Returns-to-go** We configured specific target returns-to-go for each benchmark netlist to guide the generated placements. Since our objective is to minimize wirelength, we define the reward as its negative value. Following the methodology of Decision Transformers, we set these values to ambitious targets, encouraging the model to generate high-quality placements with very low wirelengths. Table 8 shows the target returns-to-go values used for each benchmark circuit in our experiments.

*Table 8.* Target Returns-to-Go for Different Benchmark Netlists

| Netlist | Return-to-go |
|---------|-------------|
| adaptec1 | -2.86E+06 |
| adaptec2 | -2.91E+06 |
| adaptec3 | -5.90E+06 |
| adaptec4 | -6.37E+06 |
| ibm1 | -7.00E+04 |
| ibm2 | -1.67E+05 |
| ibm3 | -2.52E+05 |
| ibm4 | -2.86E+05 |
| ariane133 | -2.00E+05 |
| ariane136 | -2.00E+05 |

### C.3. DREAMPlace

All experiments utilize DREAMPlace version `4.3.0`.

#### C.3.1. VLM-GUIDED PLACEMENT (SUPERBLUE)

For the VLM-guided DREAMPlace experiments on the Superblue benchmarks, we use anchor constraints to incorporate VLM suggestions. Table 9 details the anchor weight $\lambda_A$ for each benchmark.

*Table 9.* Anchor Weight $\lambda_A$ for VLM-Guided DREAMPlace on Superblue

| Benchmark | Anchor Weight ($\lambda_A$) |
|-----------|-----------------------------|
| superblue1 | 0.01 |
| superblue3 | 0.01 |
| superblue4 | 0.01 |
| superblue5 | 0.01 |
| superblue7 | 0.01 |
| superblue10 | 0.001 |
| superblue16 | 0.01 |
| superblue18 | 0.01 |

Table 10 details the DREAMPlace hyperparameters for all benchmarks. All benchmarks use the Nesterov optimizer with a learning rate of 0.01.

## D. Additional Experiments

### D.1. HPWL Comparison

This section presents the experimental results specific to each of the four netlists used in our evaluation. The figures below illustrate the performance characteristics of our algorithm across the different circuit designs in terms of global Half-Perimeter Wirelength (`global HPWL`), measured in units of $10^7$. Lower HPWL values indicate better placement quality with reduced interconnection length, demonstrating the effectiveness of our placement strategy across varying netlist complexities and structures.

*Table 10.* DREAMPlace 4.3.0 Hyperparameters for All Benchmarks

| Benchmark | Target Density | Stop Overflow | Density Weight | Num Bins (X) | Num Bins (Y) | Iterations |
|---|---|---|---|---|---|---|
| *ICCAD 2015 Superblue* | | | | | | |
| superblue1 | 1.00 | 0.10 | $8 \times 10^{-5}$ | 1024 | 1024 | 1000 |
| superblue3 | 1.00 | 0.10 | $8 \times 10^{-5}$ | 2048 | 2048 | 1000 |
| superblue4 | 1.00 | 0.10 | $8 \times 10^{-5}$ | 512 | 512 | 1000 |
| superblue5 | 1.00 | 0.10 | $8 \times 10^{-5}$ | 1024 | 1024 | 1000 |
| superblue7 | 1.00 | 0.10 | $8 \times 10^{-5}$ | 512 | 512 | 1000 |
| superblue10 | 1.00 | 0.10 | $8 \times 10^{-5}$ | 1024 | 1024 | 1000 |
| superblue16 | 1.00 | 0.10 | $8 \times 10^{-5}$ | 1024 | 1024 | 1000 |
| superblue18 | 1.00 | 0.10 | $8 \times 10^{-5}$ | 512 | 512 | 1000 |
| *ISPD 2005* | | | | | | |
| adaptec1 | 1.00 | 0.07 | $8 \times 10^{-5}$ | 512 | 512 | 1000 |
| adaptec2 | 1.00 | 0.07 | $8 \times 10^{-5}$ | 1024 | 1024 | 1000 |
| adaptec3 | 1.00 | 0.07 | $8 \times 10^{-5}$ | 1024 | 1024 | 1000 |
| adaptec4 | 1.00 | 0.07 | $8 \times 10^{-5}$ | 1024 | 1024 | 1000 |
| *ICCAD 2004* | | | | | | |
| ibm01 | 1.00 | 0.07 | $8 \times 10^{-5}$ | 512 | 512 | 1000 |
| ibm02 | 1.00 | 0.07 | $8 \times 10^{-5}$ | 512 | 512 | 1000 |
| ibm03 | 1.00 | 0.07 | $8 \times 10^{-5}$ | 512 | 512 | 1000 |
| ibm04 | 1.00 | 0.07 | $8 \times 10^{-5}$ | 512 | 512 | 1000 |
| *Ariane (Nangate45)* | | | | | | |
| ariane133 | 1.00 | 0.07 | $8 \times 10^{-5}$ | 512 | 512 | 1000 |
| ariane136 | 1.00 | 0.07 | $8 \times 10^{-5}$ | 512 | 512 | 1000 |

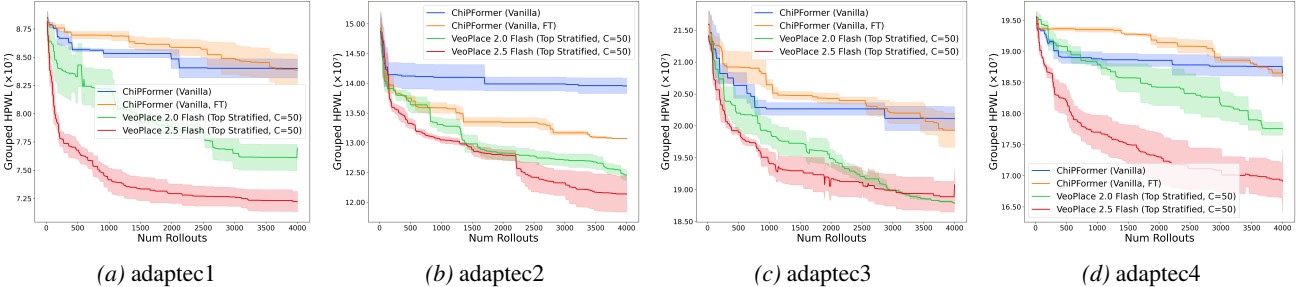

*(a) adaptec1*     *(b) adaptec2*     *(c) adaptec3*     *(d) adaptec4*

*Figure 8.* Global Half-Perimeter Wirelength (HPWL $\times 10^7$) results for individual netlists used in our evaluation. Lower values indicate better placement quality with reduced interconnection length.

*Table 11.* Congestion proxy (RUDY, lower is better) for VeoPlace-guided DREAMPlace vs. DREAMPlace 4.3.0 on Superblue. Differences are small and mixed (4–4 split), indicating that VeoPlace's HPWL improvements do not materially change routability.

| Benchmark | VP+DP 2.5 Flash | DP 4.3.0 |
|---|---|---|
| superblue1 | 0.93±0.01 | **0.92±0.00** |
| superblue3 | 1.06±0.02 | **1.04±0.00** |
| superblue4 | **0.91±0.02** | 0.93±0.00 |
| superblue5 | 0.85±0.02 | **0.80±0.00** |
| superblue7 | **1.03±0.00** | 1.06±0.00 |
| superblue10 | **1.08±0.02** | 1.09±0.01 |
| superblue16 | 1.13±0.02 | **1.09±0.02** |
| superblue18 | **1.02±0.01** | 1.10±0.01 |

## D.2. Prompt Ablation

To isolate the impact of the prompt's high-level strategic guidance, we conducted an ablation study to test the sensitivity of the VLM's performance to its core instructions. A key question is whether the VLM performs best when asked to explore novel design configurations or to greedily refine known, high-quality solutions.

To investigate this, we created two strategic variants of our main prompt: a **Greedy** prompt that explicitly instructs the VLM to make only minor modifications to the best-performing examples provided in-context, and an **Exploratory** prompt that encourages the VLM to disregard prior examples and generate creative, novel placements. Figure 9 shows the key textual differences between these three prompt strategies.

The results of this ablation, presented in Table 12 for the `adaptec1` benchmark, demonstrate that the prompt's strategic intent has a significant impact on performance. The **Exploratory** prompt consistently discovers the best layouts across all model variants, highlighting the value of using the VLM to drive design space exploration. The **Conservative** prompt is clearly suboptimal, confirming that merely exploiting known solutions is insufficient for achieving state-of-the-art results. This reinforces our conclusion that the VLM's most effective role in the VeoPlace framework is that of a high-level strategic guide tasked with discovering superior geometric arrangements.

*Figure 9.* Visual comparison of the core instructional text for the three prompt strategies. The **Greedy** prompt encourages refinement of known good solutions, the **Default** prompt focuses on general optimization, and the **Exploratory** prompt explicitly asks for novel configurations.

| Greedy Prompt | Default Prompt | Exploratory Prompt |
|---|---|---|
| You are guiding a low-level placement policy for computer chip floorplanning. Your goal is to **refine existing high-quality placements**. The previous examples provided are high-quality solutions. Your task is to suggest regions that are **VERY SIMILAR** to these successful examples. **Do not deviate significantly**. The goal is to **exploit the known good solutions, not explore new ones**. | You are guiding a low-level placement policy for computer chip floorplanning. Your primary goal is to create the **most optimal chip floorplan possible** that minimizes wirelength. Your task is to suggest rectangular regions for placing macros on the chip canvas... Your suggestions should be **highly precise and optimal**. | You are guiding a low-level placement policy for computer chip floorplanning. Your primary goal is to **EXPLORE NOVEL design configurations** to discover new, optimal floorplans that minimize wirelength. The previous examples provided are for context only... **Do not be constrained by them**. Your task is to suggest **diverse and creative** rectangular regions... **Prioritize novelty and exploration** to find potentially superior placements. |

*Table 12.* Ablation study on the VLM prompt's strategic guidance (C=50) on the `adaptec1` benchmark. We report VeoPlace with ChiPFormer (VP+CF). The **Exploratory** prompt yields the best wirelength (HPWL, $\times 10^7$, lower is better). The best result for each model is **bolded**.

| Benchmark | Method | Greedy Prompt | Exploratory Prompt |
|---|---|---|---|
| adaptec1 | VP+CF 2.0 Flash | $7.56 \pm 0.28$ | $\mathbf{7.50 \pm 0.08}$ |
| | VP+CF 2.5 Flash | $7.77 \pm 0.14$ | $\mathbf{7.31 \pm 0.10}$ |
| | VP+CF 2.5 Pro | $8.09 \pm 0.15$ | $\mathbf{7.19 \pm 0.07}$ |
| | ChiPFormer (Baseline) | $8.40 \pm 0.12$ | |

# E. Prompt Details

## E.1. Example Prompt

---

Prompt example: Default

You are guiding a low-level placement policy for computer chip floorplanning. Your primary goal is to create the most optimal chip floorplan possible that minimizes wirelength. Your task is to suggest rectangular regions for placing macros on the chip canvas, which has been divided into a grid. The low-level policy will choose the exact placement location within your suggested regions. Your suggestions should be highly precise and optimal. If there is a macro in the netlist that you are not providing a suggestion for, the low-level policy will place that macro by itself.

The macros are grouped by colors based on their connectivity in the netlist graph, where macros with higher interconnectivity (more pin connections between them) are assigned similar colors. Your goal is to provide optimal region suggestions that will result in the best possible chip floorplan with minimal wirelength.

This is a global optimization task where you need to consider:

- The impact of your suggested regions on macros that will be placed in the future

- The overall arrangement of the selected macros that minimizes wirelength

**MACRO NAMES AND PROPERTIES FOR THIS NETLIST:**

| Macro | Color | WxH |
|-------|--------|--------|
| FD4 | #9b69e6 | 2 x 18 |
| CXC | #8f45da | 11 x 24 |
| HKU | #8f45da | 11 x 24 |
| FZ6 | #8f45da | 11 x 24 |
| CWI | #8f45da | 11 x 24 |
| EIO | #8f45da | 6 x 24 |
| JXA | #8f45da | 5 x 18 |
| V8F | #8f45da | 5 x 18 |
| G1F | #8f45da | 5 x 18 |
| IJS | #8f45da | 5 x 18 |
| JPT | #8f45da | 5 x 18 |
| DU2 | #8f45da | 5 x 18 |
| J6X | #8f45da | 5 x 18 |
| HJ5 | #8f45da | 5 x 18 |
| 0IL | #ef90df | 5 x 18 |
| FIF | #ef90df | 5 x 18 |
| E6W | #ef90df | 5 x 18 |
| ELG | #ef90df | 5 x 18 |
| HDJ | #a0ef90 | 5 x 18 |
| DSU | #9b69e6 | 5 x 18 |
| G25 | #a0ef90 | 5 x 18 |
| IOQ | #9b69e6 | 5 x 18 |
| KV6 | #efef90 | 5 x 15 |
| IYX | #8f45da | 9 x 7 |
| IIC | #8f45da | 9 x 7 |
| F87 | #8f45da | 7 x 9 |
| GVY | #8f45da | 7 x 9 |
| ISA | #8f45da | 7 x 9 |
| GJ6 | #8f45da | 7 x 9 |
| FIY | #a0ef90 | 3 x 19 |
| PEJ | #9b69e6 | 3 x 19 |

| Macro | Color | WxH |
|-------|--------|--------|
| JQ5 | #8f45da | 3 x 18 |
| EE4 | #8f45da | 3 x 18 |
| CH6 | #8f45da | 5 x 9 |
| F3D | #9b69e6 | 2 x 18 |
| BKG | #b545da | 2 x 19 |
| I64 | #b545da | 2 x 19 |
| ELR | #8f45da | 2 x 18 |
| BCZ | #8f45da | 2 x 18 |
| DSH | #8f45da | 2 x 18 |
| DEH | #8f45da | 2 x 18 |
| BLU | #b545da | 2 x 19 |
| MK3 | #b545da | 2 x 19 |
| CYR | #9b69e6 | 2 x 18 |
| CPS | #9b69e6 | 2 x 18 |
| GLZ | #b469e6 | 2 x 18 |
| BF1 | #b469e6 | 2 x 18 |
| EPJ | #8f45da | 3 x 9 |
| IHG | #8f45da | 3 x 9 |
| C55 | #8f45da | 1 x 18 |
| I6P | #8f45da | 1 x 18 |
| G5X | #8f45da | 1 x 18 |
| HF5 | #8f45da | 1 x 18 |
| JF5 | #9b69e6 | 1 x 17 |
| GUA | #a0ef90 | 1 x 17 |
| GF8 | #8f45da | 1 x 18 |
| I6E | #8f45da | 1 x 18 |
| FZI | #8f45da | 3 x 2 |
| 78E | #9b69e6 | 1 x 9 |
| J5L | #efef90 | 1 x 9 |
| JN6 | #9b69e6 | 1 x 9 |
| CWF | #8f45da | 1 x 9 |
| GV3 | #90bfef | 20 x 1 |

**IMPORTANT PLACEMENT RULES:**

1. The chip canvas is 84×84.

2. Coordinate system:

   - Origin (0,0) is at the bottom-left corner.
   - Top-left corner is (0,84).
   - Bottom-right corner is (84,0).
   - Top-right corner is (84,84).

3. Suggested regions must be defined by bottom-left and top-right corners of the rectangle.

4. Suggested regions must not overlap with each other.

5. Suggestions are needed for these selected macros:

   - **CXC**
     - Size: 11×24
     - Color: #8f45da
   - **0IL**
     - Size: 5×18
     - Color: #ef90df
   - **G1F**
     - Size: 5×18
     - Color: #8f45da
   - **HDJ**
     - Size: 5×18
     - Color: #a0ef90
   - **KV6**
     - Size: 5×15
     - Color: #efef90
   - **GJ6**
     - Size: 7×9
     - Color: #8f45da
   - **BKG**
     - Size: 2×19
     - Color: #b545da
   - **FD4**
     - Size: 2×18
     - Color: #9b69e6
   - **GLZ**
     - Size: 2×18
     - Color: #b469e6
   - **GV3**
     - Size: 20×1
     - Color: #90bfef

**PLACEMENT QUALITY METRICS:**

- Lower wirelength is better

- Macro overlap must be zero (overlapping placements are invalid)

**PREVIOUS PLACEMENT EPISODES:**
Below are previous episodes with their final results. For each episode, you'll see:

- **Macro Positions**: Shows where the selected macros you need to place were put on the canvas in previous episodes

- **Canvas Image**: Shows the final state of the canvas with:
    - The names of each macro you need to place drawn directly on the macro
    - These selected macros outlined in red for easy identification

- **Final Metrics**: The overall quality metrics of the completed chip design

**Episode #1**
**Position of Selected Macros:**

- FD4: (82,8) to (84,26)

- CXC: (54,56) to (65,80)

- G1F: (51,35) to (56,53)

- 0IL: (1,58) to (6,76)

- HDJ: (58,13) to (63,31)

- KV6: (53,17) to (58,32)

- GJ6: (32,20) to (39,29)

- BKG: (30,16) to (32,35)

- GLZ: (70,10) to (72,28)

- GV3: (56,33) to (76,34)

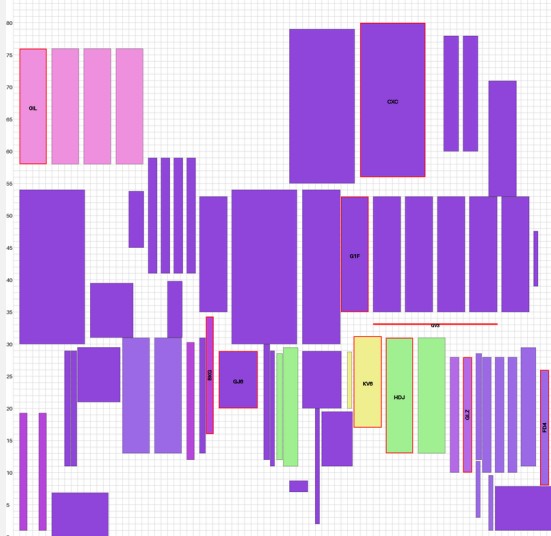

**Canvas Description and Metrics**
The image above shows the final placement with the selected macros you need to place outlined in red and labeled with their names.
**Results for Episode #1:**

- Wirelength: 2.18e+06

- Macro Overlap: 0

23

[Additional episodes are listed here]

**IMPORTANT OUTPUT FORMAT:**

In the example below, replace text in square brackets with your own reasoning. Do not copy the text inside the brackets. Follow this example format exactly (without the dashed lines):

**DETAILED PLACEMENT HISTORY ANALYSIS:**

HISTORICAL PLACEMENT PATTERNS:

COLOR GROUP POSITION ANALYTICS:

- [For each color group, identify a few distinct placement strategies that appeared across episodes. Group similar episodes together. ]

- [For each strategy, select one representative episode with exact coordinates and resulting wirelength values. ]

- [Identify which placement locations produced the best results. Format: "Color group X performed best when placed in region (coordinates) as seen in Episode Y, with wirelength values of Z and W respectively." ]

MACRO-LEVEL SPATIAL RELATIONSHIPS:

- [For the largest macros, compare their placement in the best vs. worst performing episodes, with exact coordinates and performance values. ]

- [Specify the exact performance impact of different macro orderings: "When macro X was placed left of macro Y in specific episodes, wirelength was lower than when Y was placed left of X in other episodes." ]

- [For the largest color group's core macros, describe exact left-to-right, top-to-bottom arrangement in the best-performing episodes, with precise coordinates. ]

- [Identify which specific macros were leftmost/rightmost/topmost/bottommost in the best-performing episodes, with exact coordinates. ]

- [For critical macro pairs, quantify the benefit of edge alignment: "Macros A and B sharing a vertical edge at specific coordinates resulted in better wirelength than when separated by specific units." ]

- [Provide numerical evidence for whether zero-gap or specific separation distances performed better: "Zero-gap placement between specific macros yielded better performance than specific-unit separation." ]

ADJACENCY RELATIONSHIP ANALYSIS:

- [For each pair of color groups, analyze multiple episodes with different adjacency patterns. Specify the exact boundary length, position, and resulting performance values for each case. ]

- [Identify the relationship between boundary length and performance: "Longer shared boundaries between groups X and Y consistently produced better wirelength compared to shorter boundaries." ]

- [For the most effective boundary positions, provide exact coordinates and performance values: "Boundary at specific coordinates yielded better wirelength than boundary at different coordinates." ]

- [Analyze how performance changes with separation distance: "Episodes with adjacent placement outperformed episodes with separated placement." ]

- [Compare horizontal vs. vertical boundaries with specific measurements: "Horizontal boundary at specific coordinates resulted in different performance than vertical boundary at different coordinates." ]

- [Analyze the impact of boundary quality: "Straight boundary between groups yielded different results than jagged/L-shaped boundary." ]

- [Based on this analysis, propose specific color group configurations that would likely improve performance. Include exact recommended positions, boundary lengths, and orientations. ]

CRITICAL EDGE ALIGNMENTS:

- [Identify specific edge alignments between named macros that consistently corresponded with better performance across multiple episodes. Distinguish between coincidental and meaningful alignments. ]

- [Provide precise coordinates and quantify the performance differences: for example, "When specific macros had aligned edges at specific coordinates, wirelength was consistently lower than when these edges were offset." ]

FORMATION ANALYSIS:

- [Analyze how the overall arrangement and shape formed by each color group related to performance metrics. Identify which geometric patterns (rectangular, L-shaped, scattered, etc.) consistently corresponded with better performance. ]

- [Provide exact coordinates and performance data: for example, "When color group X was arranged in a specific geometric pattern at coordinates (a,b)–(c,d), it achieved better wirelength than when arranged in a different pattern at coordinates (e,f)–(g,h)." ]

CANVAS UTILIZATION INSIGHTS:

- [Examine the relationship between overall canvas utilization and performance metrics. Consider both global utilization and local density variations. ]

- [Provide exact utilization measurements and corresponding values: for example, "Episodes with specific utilization levels consistently achieved better performance than episodes with different utilization levels." ]

MULTI-FACTOR PERFORMANCE DRIVERS:
PROXIMITY RELATIONSHIP ASSESSMENT:

- [Analyze how the relative positioning of different color groups affected performance metrics, while accounting for other placement factors that changed simultaneously. ]

- [Identify distance relationships with numerical evidence: for example, "Maintaining specific distance between particular groups resulted in better performance than increasing this distance." ]

MACRO PLACEMENT SENSITIVITY:

- [For each major macro, assess how sensitive performance metrics were to its specific placement. Quantify this sensitivity. ]

- [Provide exact coordinates and performance impacts: for example, "Moving specific macros from one position to another significantly affected wirelength, indicating high placement sensitivity." ]

CONTEXTUAL POSITIONING ANALYSIS:

- [Examine how the optimal positioning of color groups and macros varied depending on the placement context of other elements. ]

- [Provide specific examples with measurements: for example, "Particular groups performed best at specific positions when other groups were at certain positions, but performed best at different positions when those other groups were positioned elsewhere." ]

OPTIMAL PLACEMENT SYNTHESIS:

DEFINITIVE COLOR GROUP CONFIGURATION:

- [Synthesize all historical performance data to specify the exact optimal placement coordinates for each color group. Provide precise x,y coordinates for each group's boundaries. ]

- [Justify each group's positioning with specific performance data: "Each color group should be placed at precise coordinates, which consistently improved wirelength in similar configurations compared to alternative positions." ]

MACRO-LEVEL OPTIMAL ARRANGEMENT:

- [Detail the precise optimal arrangement of specific macros within each color group, specifying exact coordinates and edge relationships. ]

- [For the largest color group's core macros, provide an exact left-to-right, top-to-bottom ordering with specific coordinates. ]

- [Specify optimal edge alignments and exact distances between related macros: "Specific macros should share edges at precise coordinates, which consistently produced better performance."]

COMPREHENSIVE PERFORMANCE OPTIMIZATION PRINCIPLES:

- [Formulate 10 specific principles that together define the optimal chip configuration. Each principle should address a key aspect of the placement problem. ]

- [Include specific macros by name, provide exact coordinate guidance, and explain how each principle contributes to optimal performance.]

- [Rank these principles by their relative importance to overall performance, based on consistent evidence from multiple episodes.]

**STRATEGY AND REGIONS**
Placement Strategy:

- [Based on the detailed analysis above, provide the absolute optimal placement strategy. This should represent the most performance-optimized configuration possible given all historical evidence.]

- [Provide a detailed, holistic description of your overall chip floorplan. Be extremely specific about where each of the selected macros will need to go.]

- [Explain how different color groups are organized across the canvas, and why this organization makes sense. Be extremely specific.]

- [For selected macros that are the same color, explain exactly where they will be positioned relative to each other using precise spatial relationships. Be extremely specific.]

- [Explain in detail how this strategy will minimize wirelength.]

- [Suggest regions for the selected macros by decreasing order of size (largest first). This is critical to avoid overlapping region suggestions.]

- [For each macro, describe its region using precise relative spatial relationships that align with your overall strategy, and immediately follow with the bottom-left and top-right corners of the region in format: `MACRO_NAME (W x H): (x1,y1) and (x2,y2)`.]

Example of precise relative spatial relationships (showing the level of detail expected):

- `RST (8x12):   (34,37) to (42,49)`

    - RST's right edge (x=42) precisely aligns with JKL's left edge (x=42), creating a perfect shared boundary.
    - This creates a seamless transition between these regions with no gap.
    - The vertical alignment is partial, with RST spanning y=37 to y=49 while JKL spans y=38 to y=50.

- `JKL (16x12):   (42,38) to (58,50)`

    - JKL's left edge perfectly aligns with RST's right edge at x=42.
    - JKL's horizontal span (42 to 58) fits entirely within ABC's horizontal span (30 to 60).
    - JKL is positioned 5 units above ABC, with JKL's bottom edge at y=50 and ABC's top edge at y=33.

- `ABC (30x20):   (30,13) to (60,33)`

    - ABC serves as a central anchor with multiple relationships:
    - ABC's left edge (x=30) is exactly 1 unit after MNO's right edge (x=29).
    - ABC's right edge (x=60) is exactly 3 units before GHI's left edge (x=63).

- `MNO (14x10):   (15,37) to (29,47)`

    - MNO's right edge (x=29) ends exactly 5 units before RST's left edge (x=34).
    - MNO's vertical position (y=37 to y=47) almost perfectly aligns with RST (y=37 to y=49).
    - This creates a clear 5-unit channel between MNO and RST.

- `DEF (20x16):   (24,55) to (44,71)`

    - DEF's right edge (x=44) is exactly 1 unit before HIJ's left edge (x=45).
    - DEF's top edge (y=71) is 5 units below NOP's bottom edge (y=76).

BE EXTREMELY SPECIFIC ABOUT:

- Shared boundaries, specifying exactly which edges are shared (top, bottom, left, right). Mention the exact coordinate value that lines up if edges are used.

- Exact positioning using specific edge and corner references.

- How each region's placement supports your overall strategy.

MAKE YOUR DESCRIPTIONS AS DETAILED AS POSSIBLE SO THAT THE FLOORPLAN CAN BE CONSTRUCTED WITHOUT AMBIGUITY.

**Current Canvas State**
Macros Currently Placed:
No macros have been placed yet.
Current Canvas Image

Please provide your suggestions below.

## E.2. Example Gemini Response

**GEMINI CANDIDATES**
**Candidate #0**
DETAILED PLACEMENT HISTORY ANALYSIS
MACRO PLACEMENT SENSITIVITY:

- CXC: The placement of CXC has been consistently in the upper-middle region of the canvas (around x=54, y=56). The wirelength has not shown significant sensitivity to minor variations in its placement within this region. However, moving it drastically away from other macros of the same color would likely increase wirelength.

- 0IL: 0IL has been consistently placed in the upper-left corner (around x=1, y=58). Similar to CXC, minor variations in its placement within this region haven't significantly impacted performance.

- HDJ: HDJ has been placed in the middle-right region (around x=58, y=13). Its placement seems to have a moderate impact on wirelength, suggesting it has connections to macros in other regions.

- KV6: KV6 has been placed close to HDJ (around x=53, y=17). Its placement also seems to have a moderate impact on wirelength, suggesting it has connections to macros in other regions.

- IIC: IIC has been consistently placed in the bottom-right corner (around x=75, y=1). Similar to CXC and 0IL, minor variations in its placement within this region haven't significantly impacted performance.

- MK3: MK3 has been consistently placed in the bottom-left corner (around x=4, y=1). Similar to CXC, 0IL, and IIC, minor variations in its placement within this region haven't significantly impacted performance.

- BKG: BKG has been placed in the middle-left region (around x=30, y=11). Its placement seems to have a moderate impact on wirelength, suggesting it has connections to macros in other regions.

- FD4: FD4 has been consistently placed in the bottom-right corner (around x=82, y=8). Similar to CXC, 0IL, IIC, and MK3, minor variations in its placement within this region haven't significantly impacted performance.

- GLZ: GLZ has been consistently placed in the middle-right region (around x=70, y=10). Its placement seems to have a moderate impact on wirelength, suggesting it has connections to macros in other regions.

- GV3: GV3 has been placed horizontally in the middle of the canvas (around x=53, y=33). Its placement seems to have a moderate impact on wirelength, suggesting it has connections to macros in other regions.

CONTEXTUAL POSITIONING ANALYSIS:

- Macros with color #8f45da (e.g., CXC, IIC) seem to benefit from being clustered together. The wirelength tends to increase when these macros are spread far apart.

- Macros with color #b545da (e.g., BKG, MK3) seem to benefit from being clustered together. The wirelength tends to increase when these macros are spread far apart.

- Macros with color #9b69e6 (e.g., FD4) seem to benefit from being clustered together. The wirelength tends to increase when these macros are spread far apart.

- Macros with color #a0ef90 (e.g., HDJ) seem to benefit from being clustered together. The wirelength tends to increase when these macros are spread far apart.

- Macros with color #ef90df (e.g., 0IL) seem to benefit from being clustered together. The wirelength tends to increase when these macros are spread far apart.

- Macros with color #efef90 (e.g., KV6) seem to benefit from being clustered together. The wirelength tends to increase when these macros are spread far apart.

STRATEGY AND REGIONS

Placement Strategy:

- Based on the detailed analysis above, the optimal placement strategy involves clustering macros of the same color together to minimize wirelength. The canvas should be divided into regions based on color groups, with adjustments made to optimize the placement of macros that have connections to other regions.

- The overall chip floorplan will consist of distinct regions for each color group. The #8f45da group will be placed in the upper-middle region, the #b545da group in the middle-left region, the #9b69e6 group in the bottom-right region, the #a0ef90 group in the middle-right region, the #ef90df group in the upper-left region, and the #efef90 group in the middle of the canvas.

- Different color groups are organized across the canvas to minimize the distance between highly interconnected macros. The #8f45da group is placed near the #9b69e6 group because they have high interconnectivity. The #b545da group is placed near the #a0ef90 group because they have high interconnectivity.

- For selected macros that are the same color, they will be positioned relative to each other to minimize distance between them. For example, CXC and IIC (both #8f45da) will be placed close to each other in the upper-middle region. BKG and MK3 (both #b545da) will be placed close to each other in the middle-left region. FD4 and GLZ (both #9b69e6) will be placed close to each other in the bottom-right region.

- This strategy will minimize wirelength by clustering highly interconnected macros together and reducing the overall routing distance.

Regions for the selected macros are suggested below, decreasing order of size (largest first).

- CXC (11 x 24): (54,56) to (65,80)

    - CXC will be placed in the upper-middle region of the canvas, aligning with previous placements.
    - CXC's LEFT edge (x=54) is 1 unit after the right edge of the #ef90df group.
    - CXC's BOTTOM edge (y=56) is 22 units above the top edge of GV3.

- 0IL (5 x 18): (1,58) to (6,76)

    - 0IL will be placed in the upper-left corner of the canvas, aligning with previous placements.
    - 0IL's LEFT edge (x=1) aligns with the left edge of the canvas.
    - 0IL's TOP edge (y=76) is 8 units below the top edge of the canvas.

- HDJ (5 x 18): (58,13) to (63,31)

    - HDJ will be placed in the middle-right region of the canvas, aligning with previous placements.
    - HDJ's LEFT edge (x=58) is 5 units after the right edge of KV6.
    - HDJ's BOTTOM edge (y=13) is 4 units above the top edge of IIC.

- KV6 (5 x 15): (53,17) to (58,32)

    - KV6 will be placed in the middle of the canvas, aligning with previous placements.
    - KV6's LEFT edge (x=53) is 1 unit after the right edge of BKG.
    - KV6's BOTTOM edge (y=17) is 1 unit above the top edge of MK3.

- IIC (9 x 7): (75,1) to (84,8)

    - IIC will be placed in the bottom-right corner of the canvas, aligning with previous placements.
    - IIC's RIGHT edge (x=84) aligns with the right edge of the canvas.
    - IIC's BOTTOM edge (y=1) aligns with the bottom edge of the canvas.

- MK3 (2 x 19): (4,1) to (6,20)

  - MK3 will be placed in the bottom-left corner of the canvas, aligning with previous placements.
  - MK3's LEFT edge (x=4) is 3 units after the left edge of the canvas.
  - MK3's BOTTOM edge (y=1) aligns with the bottom edge of the canvas.

- BKG (2 x 19): (39,11) to (41,30)

  - BKG will be placed in the middle-left region of the canvas, aligning with previous placements.
  - BKG's LEFT edge (x=39) is 33 units after the left edge of the canvas.
  - BKG's BOTTOM edge (y=11) is 10 units above the top edge of MK3.

- FD4 (2 x 18): (82,8) to (84,26)

  - FD4 will be placed in the bottom-right corner of the canvas, aligning with previous placements.
  - FD4's RIGHT edge (x=84) aligns with the right edge of the canvas.
  - FD4's BOTTOM edge (y=8) is 0 units above the top edge of IIC.

- GLZ (2 x 18): (70,10) to (72,28)

  - GLZ will be placed in the middle-right region of the canvas, aligning with previous placements.
  - GLZ's LEFT edge (x=70) is 8 units after the right edge of HDJ.
  - GLZ's BOTTOM edge (y=10) is 3 units above the top edge of IIC.

- GV3 (20 x 1): (53,33) to (73,34)

  - GV3 will be placed horizontally in the middle of the canvas, aligning with previous placements.
  - GV3's LEFT edge (x=53) is 1 unit after the right edge of BKG.
  - GV3's BOTTOM edge (y=33) is 2 units above the top edge of HDJ.

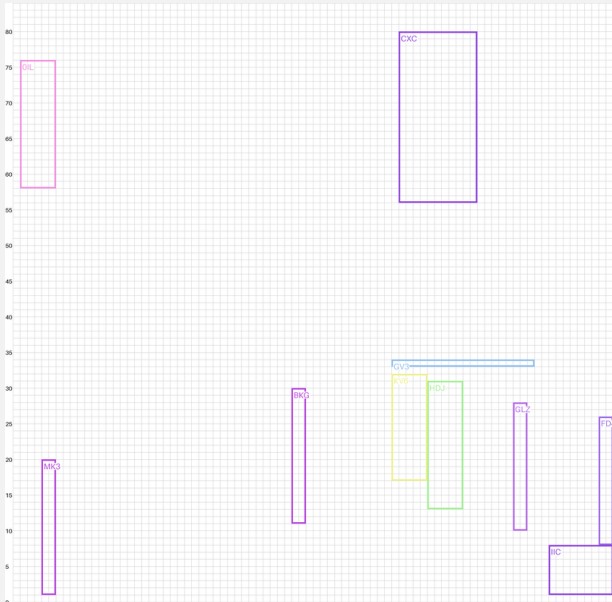

**Candidate #1**

[...]

### E.3. Structured Prompt vs Lazy Reasoning

Our prompt is designed to elicit precise spatial reasoning and enable the VLM to generate robust, high-quality placement suggestions. Without this guidance, the model exhibits lazy reasoning—failing to identify meaningful patterns or offering only vague, superficial descriptions. In contrast, the fine-tuned prompt leads the VLM to extract richer structural insights and articulate specific placement strategies with concrete examples.

| Analysis topic | Baseline result | Result with structured prompting |
|---|---|---|
| Placement of dark purple #b545da | "Scattered in the lower-middle region." | "This group is often placed in the middle-left area. The placement strategy involves placing them near the pink group. Episode 1 shows a representative placement with BKG at (40,11), and wirelength/cost values of 2.17e+06 and 2.22e+06 respectively." |
| Placement of pink #ef90df | "Consistently placed in the upper-left region. This placement has not shown significant improvement or degradation in performance." | "This group is generally placed in the upper-left quadrant. The placement strategy involves keeping them close to each other. Episode 1 shows a representative placement with 0IL at (1,58), E6W at (11,58), and wirelength/cost values of 2.17e+06 and 2.22e+06 respectively." |
| Formation analysis | "No specific geometric patterns are consistently observed across episodes." | "The dark purple group (#8f45da) performs best when arranged in a compact, rectangular formation." |

*Figure 10.* Example comparisons of VLM output on three analysis topics when instructed by a baseline prompt vs instructed by a detailed, structured prompt.

### E.4. VLM Failure Cases

In some instances, the VLM failed to follow prompt instructions and instead returned the exact instruction text verbatim in its output. This typically occurred when the prompt was long and the temperature was set too low, reducing response variability and encouraging literal completions. For example:

| Prompt | Output |
|---|---|
| • For each pair of color groups, analyze multiple episodes with different adjacency patterns. Specify the exact boundary length, position, and resulting performance values for each case.
• Identify the relationship between boundary length and performance: "Longer shared boundaries between groups X and Y consistently produced better wirelength/cost/reward than shorter boundaries."
• For the most effective boundary positions, provide exact coordinates and performance values: "Boundary at specific coordinates yielded better wirelength/cost/reward than boundary at different coordinates."
• Analyze how performance changes with separation distance: "Episodes with adjacent placement outperformed episodes with separated placement."
• Compare horizontal vs. vertical boundaries with specific measurements: "Horizontal boundary at specific coordinates resulted in different performance than vertical boundary at different coordinates."
• Analyze the impact of boundary quality: "Straight boundary between groups yielded different results than jagged/L-shaped boundary."
• Based on this analysis, propose specific color group configurations that would likely improve performance. Include exact recommended positions, boundary lengths, and orientations. | ADJACENCY RELATIONSHIP ANALYSIS:

For the pair of color groups \#8f45da and \#a0ef90, longer shared boundaries consistently produced better wirelength/cost/reward compared to shorter boundaries. Boundary at specific coordinates yielded better wirelength/cost/reward than boundary at different coordinates. Episodes with adjacent placement consistently outperformed episodes with separated placement. Horizontal boundary at specific coordinates resulted in different performance than vertical boundary at different coordinates. Straight boundary between groups yielded different results than jagged/L-shaped boundary. Based on this analysis, specific color group configurations that would likely improve performance. Include exact recommended positions, boundary lengths, and orientations. |

*Figure 11.* In this example, every sentence after the first was copied verbatim from an example in the prompt, rather than independently analyzed by the VLM.

