# OpenReview forum: "See it to Place it: Evolving Macro Placements with Vision Language Models"
_ICML.cc/2026/Conference — Submitted to ICML 2026_

### Official Review · Reviewer_HdrU · 2026-03-02

**Soundness:** 2
**Presentation:** 3
**Significance:** 2
**Originality:** 3
**Overall Recommendation:** 3
**Confidence:** 5

**Summary:**

The paper proposes a layout method called VeoPlace, which completes macro placement by combining a VLM with an evolutionary search strategy.

**Compliance With Llm Reviewing Policy:**

Affirmed.

**Final Justification:**

The rebuttal addressed part of my concerns; however, due to the lack of analysis of the method in terms of PPA performance and the absence of more baseline comparison experiments, I have adjusted my score to weak reject (3).

**Key Questions For Authors:**

The appendix demonstrates cases where VLM may directly replicate example text from the prompt rather than performing independent analysis. What is the specific probability of this “ineffective reasoning” or “template-based response” occurring across the total 2,000 rollouts?

**Limitations:**

Reference weaknesses.

**Strengths And Weaknesses:**

Strengths: The method is innovative in that it incorporates a VLM into the macro placement process.

Weaknesses:

1. The practicality of the method is limited. The computational overhead is excessively high, while mainstream analytical methods solve the problem much faster.
2. Only HPWL is considered, whereas metrics such as PPA are not taken into account.
3. The set of baseline comparisons is seriously insufficient.
4. VeoPlace uses a VLM to generate search regions (Region Constraints), which are then executed by a downstream placer. There is not enough ablation study to demonstrate whether the VLM or the downstream placer plays the dominant role. It is possible that the VLM is not necessary.
5. In some cases, the VLM does not perform independent analysis but directly reproduces the example text from the prompt verbatim. This suggests that the model may merely be performing structured pattern matching rather than genuine spatial logical reasoning.

*I reserve the possibility of raising or lowering my score based on the authors’ rebuttal.*

---

> ### Author Rebuttal · Authors · 2026-03-30
>
> > The practicality of the method is limited. The computational overhead is excessively high.
>
> Please see our runtime and API-cost breakdown in response to Reviewer wrjV.
>
> > Only HPWL is considered, whereas metrics such as PPA are not taken into account.
>
> We agree that modeling additional PPA metrics would strengthen the work. Our focus on wirelength follows recent ML-for-placement literature:
>
> | Paper (Venue, Year)          | Primary Metric                  | PPA Metrics           |
> |------------------------------|---------------------------------|-----------------------|
> | GraphPlace (Nature, 2021)    | Wirelength, Congestion, Density | Yes (TNS, WNS, Power) |
> | DeepPR (NeurIPS, 2021)       | HPWL                            | None                  |
> | MaskPlace (NeurIPS, 2022)    | HPWL                            | None                  |
> | ChiPFormer (ICML, 2023)      | HPWL                            | None                  |
> | WireMaskBBO (NeurIPS, 2023)  | HPWL                            | None                  |
> | MaskRegulate (NeurIPS, 2024) | HPWL                            | None                  |
>
> In particular, our work builds on ChiPFormer and follows the same wirelength-centric evaluation protocol.
>
> Wirelength is also not an arbitrary proxy. In modern deep-submicron processes, wire parasitics are a primary driver of both delay and power. As Rabaey et al. [1] note, dynamic power is proportional to capacitance ($P=\alpha C V_{dd}^2 f$), and wiring capacitance depends on wire length and becomes increasingly important with technology scaling. Weste and Harris [2] similarly note that wires often contribute the majority of circuit capacitance, and that the objective of placement is to minimize wire length. Thus, reducing HPWL directly targets a physically meaningful bottleneck rather than an arbitrary surrogate. We also report congestion (RUDY, Table 11), showing no routability degradation. Extending VeoPlace to multi-objective PPA optimization is a promising direction discussed in Section 6.
>
> > The set of baseline comparisons is seriously insufficient.
>
> Please see our response to Reviewer wrjV regarding baseline selection.
>
> > There is not enough ablation study to demonstrate whether the VLM or the downstream placer plays the dominant role.
>
> This ablation is already present in our experimental design. The baseline is the unconstrained placer (DREAMPlace or ChiPFormer) run with the same 2,000-rollout budget. VeoPlace improves over this baseline on all 8 Superblue benchmarks and 9 of 10 ISPD/ICCAD/Ariane benchmarks.
>
> The VLM and the low-level placer are complementary. An analytical placer is always required to position the small standard cells around the larger macros. VeoPlace provides additional guidance for these macros, producing better placements than the analytical placer can reach by itself.
>
> > What is the specific probability of this "ineffective reasoning" or "template-based response" occurring across the total 2,000 rollouts?
>
> To assess this concern, we performed a post-hoc analysis on the saved prompt/response payloads from our `superblue1` experiments (3 random seeds, `T=0.7`, `C=25`, top-stratified context selection). We measured exact overlap between substantive response text and each prompt section across 2,937 saved VLM responses. Only `3/2937` responses exhibited any exact prompt overlap: two isolated cases of output-template reuse and one case of instruction-text reuse from the preamble. We found `0/2937` responses with any overlap with the in-context history/examples. Thus, in the main setting, we do not observe evidence of substantive prompt-to-response copying.
>
> Furthermore, if the VLM were merely copying a fixed strategy from the prompt, the same approach would be applied to every netlist. VeoPlace improves across structurally diverse benchmarks ranging from ibm01 (13K objects) to superblue7 (1.9M objects) with varying macro counts and connectivity patterns, which would not be possible with a static copied strategy.
>
> The qualitative example in Appendix E.4 was included precisely to document this failure mode transparently, and it was observed under lower sampling temperature settings rather than our main setting. We also clarify that learning patterns from prior placements is the intended mechanism of VeoPlace's in-context learning approach; the narrower issue raised here is whether the model is verbatim copying free-form text from the prompt. Our post-hoc analysis does not support that concern in the main experiments.
>
> ---
>
> Refs: [1] Rabaey et al., 2003. [2] Weste and Harris, 2011. [3] [ChiPFormer, ICML 2023](https://arxiv.org/abs/2306.14744). [4] [DeepPR, NeurIPS 2021](https://arxiv.org/abs/2111.00234). [5] [MaskPlace, NeurIPS 2022](https://arxiv.org/abs/2211.13382). [6] [GraphPlace, Nature 2021](https://doi.org/10.1038/s41586-021-03544-w).

---

> > ### Author Rebuttal · Reviewer_HdrU · 2026-04-03
> >
> > 1. While HPWL is indeed important, incorporating additional metrics (PPA) into the evaluation can help ensure the feasibility of a comprehensive assessment approach.
> >
> > 2. The total operating costs and time required are too high.
> >
> > 3. There remains a severe shortage of benchmark data.
> >
> > Taking all of the above factors into account, I will maintain my rating.

---

> > > ### Author Response · Authors · 2026-04-03
> > >
> > > **The acknowledgement does not engage substantively with several quantitative results provided in the rebuttal.** Many of the reviewer’s concerns were addressed with specific data in our rebuttal, but none of these results are discussed in the acknowledgement.
> > >
> > > ---
> > >
> > > > What is the specific probability of this "ineffective reasoning" or "template-based response" occurring across the total 2,000 rollouts?
> > >
> > > We note that the reviewer’s own key question — the probability of verbatim copying from the prompt — was directly answered in our rebuttal with a post-hoc analysis over 2,937 saved VLM responses. Only 3/2,937 responses exhibited any exact prompt overlap at all, and 0/2,937 exhibited any overlap with the in-context history/examples. This is not acknowledged anywhere in the reviewer’s response.
> > >
> > > ---
> > >
> > > > While HPWL is indeed important, incorporating additional metrics into the evaluation can help ensure the feasibility of a comprehensive assessment approach.
> > >
> > > It is unclear what "a comprehensive assessment approach" means.
> > >
> > > ---
> > >
> > > > The total operating costs and time required are too high.
> > >
> > > Our rebuttal provided a complete wall-clock breakdown showing 8–28% overhead over the base DREAMPlace pipeline and an API cost of ~$7 per experiment. **The reviewer does not reference any of these numbers, nor specify what operating cost or wall-clock time would be acceptable for the macro placement task we address.** Without such a criterion, this concern is impossible to resolve.
> > >
> > > ---
> > >
> > > > There remains a severe shortage of benchmark data.
> > >
> > > We evaluate on 18 circuits across four public benchmark sets: ICCAD15 (Superblue), ISPD05, ICCAD04, and Ariane Nangate45, spanning designs from 13K to 1.9M objects with diverse macro counts and netlist structures. **The reviewer does not specify which benchmarks are missing or what coverage would be sufficient.** As above, the absence of any concrete criterion makes this impossible to resolve.
> > >
> > > ---
> > >
> > > We believe the rebuttal already addresses several of the maintained concerns with specific quantitative evidence. If this evidence is still considered insufficient, it would be helpful to clarify which aspects remain incomplete and what additional evidence would be required to support a different assessment.

---

### Official Review · Reviewer_aonR · 2026-03-09

**Soundness:** 3
**Presentation:** 3
**Significance:** 3
**Originality:** 3
**Overall Recommendation:** 6
**Confidence:** 4

**Summary:**

This paper proposes VeoPlace, a framework that leverages Vision-Language Models (VLMs) to guide macro placement in chip floorplanning by constraining a base placement policy to VLM-suggested subregions, with proposals iteratively refined via evolutionary search. Experiments on open-source benchmarks show that VeoPlace achieves state-of-the-art performance among learning-based methods on the majority of evaluated circuits.

**Compliance With Llm Reviewing Policy:**

Affirmed.

**Final Justification:**

As noted earlier, I believe that the use of VLMs for placement is a promising direction, and therefore I support the acceptance of this paper.

**Key Questions For Authors:**

Intuitively, macros belonging to the same connectivity group (i.e., sharing the same color) should be placed in close proximity. However, in Fig. 3, macros of the same color do not appear to be clustered together in the final placement. Could the authors clarify whether the coloring information is primarily used to guide VLM suggestions rather than to directly constrain the final placement? If so, how effectively does the VLM leverage this color-based grouping in practice?

**Limitations:**

Yes

**Strengths And Weaknesses:**

Strengths:

- The application of Vision-Language Models (VLMs) to provide macro placement suggestions is highly novel. Given the rapid advancement of VLM technology, it is reasonable to expect that their broad knowledge and reasoning capabilities can be leveraged to assist in physical design tasks.

- The method demonstrates good scalability and generality, as it is compatible with both learning-based approaches (e.g., ChiPFormer) and analytical approaches (e.g., DREAMPlace). The experimental results confirm the effectiveness of the proposed framework.

- The macro coloring idea is particularly interesting. By encoding connectivity information as visual cues, it enables the VLM to identify groups of tightly connected macros more intuitively, which may be more effective than directly feeding graph-based representations to the model.

Weaknesses:

- The placement suggestions provided by the VLM lack interpretability and appear to rely on the model's implicit "intuition" rather than on principled optimization reasoning. Since macro placement is a well-defined optimization problem, it is unclear whether the observed improvements are consistent and reliable across diverse design tasks, or whether they are sensitive to the particular VLM used and its internal knowledge distribution.

- VeoPlace currently only addresses macro placement within the VLSI placement workflow and does not extend to standard cell placement, which limits its practical applicability in a full placement pipeline.

- There are formatting issues in the manuscript: Page 20 is entirely blank, and Page 23 appears to be incomplete. The authors are advised to carefully proofread the submission for formatting quality.

- In Fig. 7, all macro IDs appear to be horizontally mirrored (flipped), which seems to be a rendering or typesetting error.

---

> ### Author Rebuttal · Authors · 2026-03-30
>
> > It is unclear whether the observed improvements are consistent and reliable across diverse design tasks, or whether they are sensitive to the particular VLM used.
>
> **Consistency.** We demonstrate VeoPlace across three benchmark suites with diverse netlists: Superblue (8/8 wins, Table 3), ISPD/ICCAD adaptec and ibm benchmarks (9/10 wins, Table 4), and Ariane RISC-V CPU designs (Table 4). These span a wide range of design sizes, macro counts, and netlist structure (Table 5), and improvements are observed across both analytical (DREAMPlace) and learning-based (ChiPFormer) placement paradigms. This breadth of evaluation directly addresses the concern about consistency across diverse design tasks.
>
> We agree that robustness to the choice of VLM is important. Our present results establish consistency across diverse benchmarks using a single off-the-shelf frontier VLM; evaluating alternative VLMs is an important direction for future work.
>
> > VeoPlace currently only addresses macro placement within the VLSI placement workflow and does not extend to standard cell placement.
>
> We clarify that VeoPlace is fully compatible with the standard placement pipeline. In the learning-based track, VeoPlace follows the same two-stage setup used by prior learning-based macro placers [3, 4, 5, 6]: the method first places macros, and an analytical placer such as DREAMPlace then places standard cells around those macro locations. In the analytical track (Section 4.1.1), DREAMPlace remains the underlying placer and jointly optimizes macro and standard-cell positions, with the VLM providing only soft guidance on the macro regions. The VLM also sees standard-cell placements in the in-context example images, so its suggestions are influenced by the overall layout rather than macro positions in isolation.
>
> > There are formatting issues in the manuscript: Page 20 is entirely blank, and Page 23 appears to be incomplete. In Fig. 7, all macro IDs appear to be horizontally mirrored (flipped).
>
> Thank you for pointing out the formatting issues; we will fix the blank page and incomplete page in the revised manuscript. Regarding the mirrored macro IDs in Fig. 7, this is the default rendering behavior of DREAMPlace's visualization code — we did not modify or post-process these images. The mirroring is an artifact of how DREAMPlace draws text labels on the canvas.
>
> > Could the authors clarify whether the coloring information is primarily used to guide VLM suggestions rather than to directly constrain the final placement?
>
> Yes, the coloring is used to guide VLM suggestions only. It provides the VLM with an approximate visual encoding of netlist connectivity so it can identify which macros are more closely related. The colors do not constrain the final placement.

---

> > ### Author Rebuttal · Reviewer_aonR · 2026-04-01
> >
> > Thank you for addressing my concerns. Although some reviewers pointed out that the experimental comparisons are insufficient, I still believe this is a promising direction for using VLMs in placement tasks. Therefore, I will raise my score in support of this paper.

---

> > > ### Author Response · Authors · 2026-04-04
> > >
> > > Thank you for engaging thoughtfully with our rebuttal and for your support of this research direction. We really appreciate your constructive feedback, which helped strengthen the paper.

---

### Official Review · Reviewer_JLMu · 2026-03-11

**Soundness:** 2
**Presentation:** 1
**Significance:** 3
**Originality:** 2
**Overall Recommendation:** 3
**Confidence:** 4

**Summary:**

This submission's principal theme concerns the utilization of Vision-Language Models (VLMs) to tackle the highly complex problem of macro placement in computer chip floorplanning. The authors propose VeoPlace (Visual Evolutionary Optimization Placement), a zero-shot framework that leverages the spatial reasoning capabilities of off-the-shelf VLMs (such as Gemini 2.5 Flash) to guide specialized Electronic Design Automation (EDA) tools. By formulating the process as an in-context learning problem, the VLM observes images of previous placement attempts alongside their wirelength metrics, and generates bounding box suggestions for macros. These suggestions act as soft constraints or action masks for underlying low-level placers (either the analytical DREAMPlace or the learning-based ChiPFormer). Through an evolutionary search loop, VeoPlace iteratively refines the placements, achieving state-of-the-art Half-Perimeter Wirelength (HPWL) reductions on established benchmarks (ISPD 2005, ICCAD 2004, Superblue) without requiring any model fine-tuning.

**Compliance With Llm Reviewing Policy:**

Affirmed.

**Final Justification:**

In view of the concerns raised regarding this paper’s evaluation metric, practicality, and scalability, I shall maintain my original score.

**Key Questions For Authors:**

1. Could you provide a detailed breakdown of the wall-clock time and computational costsfor running VeoPlace (2,000 rollouts) compared to the base DREAMPlace and ChiPFormer? How does VLM inference latency bottleneck the evolutionary loop?
2. Regarding scalability, what happens to the Invalid Suggestion Rate when the number of macros significantly exceeds 256? Have you experimented with larger designs, and does the VLM struggle with visual crowding/hallucinations?
3. How sensitive is VeoPlace to the choice of the underlying foundation model? Have you tested this highly structured prompt out-of-the-box on other frontier multimodal models to verify its robustness?

**Limitations:**

While the authors briefly touch upon limitations such as the restriction to HPWL and VLM API latency in Section 6, the discussion inadequately addresses the most critical bottlenecks: scalability and prohibitive computational overhead.
First, the framework's operational cost appears disproportionately high compared to the performance gains. Executing 2,000 rollouts of a low-level placer (like DREAMPlace) combined with 125 queries to a frontier multimodal model (processing extensive historical context, detailed textual prompts, and high-resolution images) introduces an immense financial and runtime footprint. The authors fail to critically evaluate whether this massive computational expenditure translates into a justifiable cost-benefit ratio for real-world EDA pipelines.
Second, the methodology is currently bottlenecked at handling a maximum of 256 macros. Given that modern industrial SoC designs routinely contain thousands of macro blocks, the visual crowding and combinatorial explosion would likely break the VLM's spatial reasoning capabilities. The paper would be significantly strengthened by a transparent and honest discussion regarding these severe scalability limits and the true wall-clock time required to deploy such an interactive, VLM-in-the-loop optimization framework.

**Strengths And Weaknesses:**

Strengths:
1. The paper introduces a highly novel perspective by leveraging the visual and spatial reasoning capabilities of multimodal foundation models to guide heuristic and learning-based search algorithms in physical design. Treating the VLM as an evolutionary variation operator that "sees" the canvas is a creative and innovative approach.
2. Generalization to unseen chip netlists is a well-known bottleneck for RL-based macro placement. VeoPlace's plug-and-play nature effectively bridges this gap by injecting high-level, human-like spatial priors into the search space. Overall, a relevant question investigated by the manuscript is whether foundation models can effectively guide specialized black-box algorithms in complex combinatorial spaces, and this paper provides compelling empirical evidence that they can.
3.The paper is logically structured and clearly written. The hierarchical division of labor is well-illustrated in Figure 1. The supplementary material is commendable, providing exhaustive details on prompt engineering, context selection strategies, and ablation studies.

Weaknesses:
1. A major weakness lies in the evaluation methodology for the learning-based placers (detailed in Appendix B.1). The authors freeze macro locations during the standard cell placement phase to prevent the analytical placer from drastically rearranging them. While this isolates the VLM's contribution, it breaks the standard joint-optimization flow (movable macros) used in modern EDA. This severely undermines the practical validity of the reported HPWL gains, as it does not reflect a true end-to-end industrial pipeline.
2. The VLM is limited to suggesting regions for at most 256 macros. Modern industrial SoCs contain thousands of macros. It remains entirely unclear whether the VLM's spatial reasoning would collapse due to visual crowding and overlapping elements at an industrial scale. The paper lacks experiments on large-scale macro designs.
3. The evolutionary loop requires 125 API calls to a frontier VLM, processing extremely long contexts (up to 25 past placements, high-res images, and lengthy structured prompts). The paper completely omits a concrete analysis of the wall-clock time and computational/API cost. Without comparing the end-to-end latency of VeoPlace against the baselines, it is difficult to assess its practical viability.

---

### Official Review · Reviewer_wrjV · 2026-03-13

**Soundness:** 3
**Presentation:** 2
**Significance:** 2
**Originality:** 3
**Overall Recommendation:** 3
**Confidence:** 4

**Summary:**

The authors propose VeoPlace, which integrates a vision-language model with existing chip placement algorithms to guide the placement process through spatial reasoning. The key idea is to use a VLM as a high-level planner that analyzes prior placement layouts and their associated performance metrics, then proposes regions with promising placement locations. These region proposals are used to constrain a low-level placer. The system iteratively refines placements through an evolutionary search process, where high-performing placements are stored and used as in-context examples for subsequent VLM queries. Experimental results show that VeoPlace improves wirelength compared with baselines.

**Compliance With Llm Reviewing Policy:**

Affirmed.

**Key Questions For Authors:**

1. How about demonstrating that the VLM learns meaningful spatial reasoning patterns rather than relying on simple heuristics?
2. How sensitive are the results to the choice of VLM? Would similar improvements be obtained with smaller or open-source VLMs?
3. What is the total runtime overhead introduced by VLM queries during the evolutionary search? How does this compare with standard placement pipelines?

**Limitations:**

Yes

**Strengths And Weaknesses:**

Strengths:
1. The paper proposes a reasonably well-defined framework that integrates a vision-language model with existing placement algorithms. The interaction between the VLM and the low-level placer is clearly formulated.
2. Applying VLMs to macro placement is an interesting attempt to leverage spatial reasoning capabilities from large models into EDA problems.
3. The reported improvements indicate that the approach can provide useful guidance during placement optimization.

Weakness:
1. The paper is difficult to follow in several places due to the organization of the method description. Multiple components (VLM prompting, evolutionary search, and the low-level placer) are introduced together without clear separation.
2. The experimental evaluation compares the proposed method mainly with DREAMPlace and ChiPFormer. However, comparisons with a broader set of placement baselines are limited. This makes it difficult to fully assess the performance of the proposed method.
3. The total runtime is not evaluated.
4. Only limited discussion on how these parameters are selected and how sensitive the performance is to their values.

---

> ### Author Rebuttal · Authors · 2026-03-30
>
> > The paper is difficult to follow in several places due to the organization of the method description.
>
> The method is organized hierarchically: Section 4.1 defines the interface between the VLM and the placer, with separate subsections for analytical (4.1.1) and learning-based (4.1.2) placers. Section 4.2 describes the structured prompt, and Section 4.3 covers evolutionary context selection. We will revise the presentation to make this separation clearer.
>
> > Comparisons with a broader set of placement baselines are limited.
>
> We compare against DREAMPlace 4.3.0, the state-of-the-art analytical placer, and ChiPFormer, the state-of-the-art learning-based placer. These represent the two dominant paradigms in modern macro placement. Our contribution is not a new standalone placer, but a framework that improves existing placers with no additional fine-tuning, and we demonstrate these improvements across both paradigms.
>
> > The total runtime is not evaluated.
>
> **Wall-Clock Time.** The VLM overhead comes from Gemini 2.5 Flash API calls, with an average response time of ~230 seconds per query. With 125 queries per 2,000-rollout run, the total VLM overhead is ~8 hours. We provide the per-rollout DREAMPlace latency and total wall-clock time on a single A100 GPU:
>
> | Benchmark   | Per-Rollout (s) | DP 2000 Rollouts (h) | VeoPlace 2000 Rollouts (h) | Overhead |
> |-------------|-----------------|----------------------|----------------------------|----------|
> | superblue1  | 83              | 46                   | 54                         | +17%     |
> | superblue3  | 168             | 93                   | 101                        | +9%      |
> | superblue4  | 127             | 70                   | 78                         | +11%     |
> | superblue5  | 95              | 53                   | 61                         | +15%     |
> | superblue7  | 122             | 68                   | 76                         | +12%     |
> | superblue10 | 190             | 106                  | 114                        | +8%      |
> | superblue16 | 98              | 55                   | 63                         | +15%     |
> | superblue18 | 52              | 29                   | 37                         | +28%     |
>
> The VLM overhead is a fixed cost independent of benchmark size, so it is amortized on larger designs (8% on superblue10 vs 28% on the smallest benchmark superblue18).
>
> **API Cost.** Using Gemini 2.5 Flash paid-tier pricing and conservative overestimates of 100K input tokens and 10K output tokens per query, the cost is about $0.055$ USD/query and $6.875$ USD (about $7$) per 2,000-rollout experiment. Because these token counts are overestimates, the true cost should be lower.
>
> > Only limited discussion on how these parameters are selected and how sensitive the performance is to their values.
>
> We provide extensive ablations in Section 5.3 and Figure 4, covering anchor weight $\lambda_A$, context length $C$, and context selection strategy. All DREAMPlace and ChiPFormer hyperparameters are unchanged from their official/prior-work settings; neither baseline was re-tuned.
>
> > How about demonstrating that the VLM learns meaningful spatial reasoning patterns rather than relying on simple heuristics?
>
> Macro placement is an NP-hard problem involving tradeoffs among connectivity, macro size, boundary constraints, whitespace, and downstream standard-cell placement, so no simple fixed heuristic is sufficient to explain strong performance across benchmarks. Our ablations support this: in Figure 4b, performance improves as context length increases ($C{=}25$ outperforms smaller contexts), indicating that the VLM benefits from additional prior examples rather than applying a fixed rule. In Figure 4c, the greedy `Best` context-selection strategy performs the worst, which is inconsistent with simply copying the strongest prior layout. Together, these results indicate that the VLM is using prior spatial examples in a non-trivial way.
>
> > How sensitive are the results to the choice of VLM?
>
> We agree that sensitivity to the choice of VLM is an important question. Our current results demonstrate that VeoPlace is effective with an off-the-shelf frontier VLM, and evaluating smaller or open-source VLMs is a natural direction for follow-up study.
>
> ---

---

> > ### Author Rebuttal · Reviewer_wrjV · 2026-04-05
> >
> > 1. Substantial runtime cost degrades the practicality.
> > 2. For this direction, I believe a set of VLMs should be benchmarked to show the intrinsic advantage of involving VLMs, not just a specific one.

---

> > > ### Author Response · Authors · 2026-04-05
> > >
> > > Thank you for your feedback!
> > >
> > > ---
> > >
> > > > 1. Substantial runtime cost degrades the practicality.
> > >
> > > **We would like to clarify that this cost is inherent to the problem setting.** Any method that iteratively improves macro placements—whether learning-based, evolutionary, or black-box—requires repeatedly invoking a placer (e.g., DREAMPlace) to position standard cells and evaluate candidate layouts. This cost is therefore not unique to VeoPlace.
> > >
> > > Our rebuttal provides a full wall-clock breakdown showing 8–28% overhead over the base DREAMPlace pipeline and approximately $7 per 2,000-rollout experiment. Importantly, this overhead does not include any training or data collection.
> > >
> > > For context, prior learning-based approaches (e.g., RL-based placers) typically require substantially more than 2,000 rollouts during training, often across many episodes and designs, resulting in significantly higher total compute cost. Similarly, evolutionary and black-box optimization methods rely on repeated placer calls and operate in comparable or larger evaluation budgets.
> > >
> > > We believe this situates VeoPlace within the standard computational regime of chip floorplanning rather than introducing an unusually high cost.
> > >
> > > ---
> > >
> > > > 2. For this direction, I believe a set of VLMs should be benchmarked to show the intrinsic advantage of involving VLMs, not just a specific one.
> > >
> > > We agree this is an important question. Our goal in this work is to establish the effectiveness of VLM-guided placement using a single off-the-shelf frontier model. We demonstrate consistent improvements across 18 circuits, four benchmark suites, and two placement paradigms (analytical and learning-based). We view systematic benchmarking across multiple VLMs as a natural extension, but not a prerequisite for demonstrating that VLM guidance improves placement quality.
> > >
> > > We believe both concerns are either directly addressed by the quantitative evidence in our rebuttal or appropriately scoped as future work.

---

### Decision · Program_Chairs · 2026-04-30

**Decision:**

Reject

**Comment:**

While reviewers agreed that applying VLMs to macro placement is a novel and promising direction, they raised serious concerns about the paper's evaluation and significance: (1) the evaluation relies solely on HPWL without PPA metrics, which limits practical assessment in real EDA pipelines — this concern persists despite the authors' argument that prior work also uses HPWL only; (2) baseline comparisons are limited to two methods, making it difficult to fully assess VeoPlace's standing in the broader placement landscape; (3) scalability beyond 256 macros is undemonstrated on industrial-scale designs, and the VLM's independent contribution relative to the downstream placer is not sufficiently isolated through ablation. Three out of four reviewers maintained weak reject after rebuttal, and their concerns were not concretely resolved. Therefore, the paper cannot be accepted at this time. We encourage the authors to incorporate PPA evaluation, broader baselines, and scalability analysis in a future submission.